# Inequalities of visceral leishmaniasis case-fatality in Brazil: A multilevel modeling considering space, time, individual and contextual factors

**Gláucia Cota** [1]*, **Astrid Christine Erber** [2], **Eva Schernhammer** [2], **Taynãna Cesar Simões** [1]

**1** Pesquisa Clínica e Políticas Públicas em Doenças Infecto-Parasitárias, Instituto René Rachou, Fundação Oswaldo Cruz, Belo Horizonte, Brazil, **2** Department of Epidemiology, Center for Public Health, Medical University of Vienna, Vienna, Austria

* glaucia.cota@fiocruz.br

**Data Availability Statement:** Data cannot be shared publicly because there are individual information about VL cases reported in Brazil. Data

## Abstract

### Background

In Brazil, case-fatality from visceral leishmaniasis (VL) is high and characterized by wide differences between the various political-economic units, the federated units (FUs). This study was designed to investigate the association between factors at the both FU and individual levels with the risk of dying from VL, after analysing the temporal trend and the spatial dependency for VL case-fatality.

### Methodology

The analysis was based on individual and aggregated data of the Reportable Disease Information System-SINAN (Brazilian Ministry of Health). The temporal and spatial distributions of the VL case-fatality between 2007 and 2017 (27 FUs as unit of analysis) were considered together with the individual characteristics and many other variables at the FU level (socio-economic, demographic, access to health and epidemiological indicators) in a mixed effects models or multilevel modeling, assuming a binomial outcome distribution (death from VL).

### Findings

A linear increasing temporal tendency (4%/year) for VL case-fatality was observed between 2007 and 2017. There was no similarity between the case-fatality rates of neighboring FUs (non-significant spatial term), although these rates were heterogeneous in this spatial scale of analysis. In addition to the known individual risk factors age, female gender, disease's severity, bacterial co-infection and disease duration, low level schooling and unavailability of emergency beds and health professionals (the last two only in univariate analysis) were identified as possibly related to VL death risk. Lower VL incidence was also associated to VL case-fatality, suggesting that unfamiliarity with the disease may delay appropriate medical management: VL patients with fatal outcome were notified and had VL treatment started

were provided by the Brazilian Minister of Health after Ethics Committee approval and the establishing a commitment to maintaining the confidentiality of personal information. Access to data of the Brazilian health surveillance system can be request through the Integrated Access to Information Platform (https://esic.cgu.gov.br/sistema/site/index.aspx) upon presentation of a research project and the ethic commitee approval.

**Funding:** GC is currently receiving a grant [301384/2019] from National Counsel of Technological and Scientific Development (CNPq). The funder had no role in study design, data collection and analysis, decision to publish, or preparation of the manuscript.

**Competing interests:** The authors have declared that no competing interests exist.

6 and 3 days later, respectively, in relation to VL cured patients. Access to garbage collection, marker of social and economic development, seems to be protective against the risk of dying from VL. Part of the observed VL case-fatality variability in Brazil could not be explained by the studied variables, suggesting that factors linked to the intra FU environment may be involved.

## Conclusions

This study aimed to identify epidemiological conditions and others related to access to the health system possibly linked to VL case-fatality, pointing out new prognostic determinants subject to intervention.

## Author summary

Visceral leishmaniasis (VL) is a potentially fatal disease if not diagnosed and treated promptly. The VL case-fatality in Brazil is the highest rate in the world, reaching an average of 7% and in some regions, more than 15%. In the last years, some improvements in the VL approach have been reached in Brazil, such as the widespread use of rapid diagnostic tests and liposomal amphotericin B for treatment of selected high risk of death cases. Despite these interventions, increase in case-fatality rates were observed. In this study we explored the factors related to the case-fatality from VL using a mixed modeling that encompasses different intervening factors such as time/spatial trends and factors linked to the individual and socio-economic indicators. For the first time, factors unrelated to the patients' clinical condition emerge as possibly related to VL case-fatality, such as low educational level, unavailability of emergency beds and health professionals, suggesting the harmful influence of conditions of limited access to health services. In addition to these significant effects observed in the spatial scale of analysis, this study points to the influence of contextual factors linked to each geopolitical unit. The determinants of death among VL cases may differ according to the region, which requires specific actions planned locally, including increased access to health system qualified to recognize and properly treat VL.

## Introduction

Visceral leishmaniasis (VL), also known as kala-azar, is a neglected tropical disease endemic in more than 65 countries, caused by *Leishmania donovani* and *L. infantum* (synonym *L. chagasi*) and transmitted by sandflies. Of the 200000 to 400000 new cases annually of VL worldwide, more than 90% occur in six countries: India, Bangladesh, Sudan, South Sudan, Ethiopia and Brazil [1]. Visceral leishmaniasis is a potentially fatal disease if not diagnosed and treated promptly [2]. VL case-fatality is one of the highest among all neglected infectious diseases, reaching 7% in Brazil, the highest rate in the world [3]. From the perspective of global disease burden, also in comparison with other neglected tropical diseases, leishmaniasis is the third infection with the highest number of accumulated deaths [4].

Substantial progress has been made in the elimination of VL in all world, mainly in Southeast Asia—Bangladesh, India, and Nepal [5]. In these three countries, significant progress has been noted in VL incidence due to efforts focused on vector control and improved surveillance

to reduce transmission [6], but also in the reduction of mortality, probably due to the expansion of access to diagnosis and treatment [7]. Treatment options for VL have moved from a reliance on antimonial monotherapy to new alternatives, including different lipidic formulations of amphotericin B, the oral drug miltefosine, and injectable paromomycin, besides different combinations of drugs [8].

In the Americas region, the estimated annual incidence of VL is 4500 to 6800 cases; of these, 4200 to 6500 cases (>95%) occurred in Brazil alone [9], where the VL incidence is relatively stable. In contrast, the case-fatality is continuously rising in Brazil, despite the several advances implemented in the VL approach in recent years. In particular, two interventions were instituted with the aim of reducing the VL case-fatality without the expected impact: the incorporation of rapid tests for the diagnosis and the expansion of the criteria for liposomal amphotericin B use, a drug with a more favorable safety profile.

Although several studies have already identified factors associated with the risk of death from VL, the high case-fatality rates in some of the more developed Brazilian urban cities, remains an unresolved issue. Until now, most of the predictors of fatal outcome among VL patients confirmed in the literature are expression of the severity of the disease itself, more than true prognostic markers, including malnutrition, thrombocytopenia, leukopenia, renal failure (creatinine >1.5 mg/dl), diarrhea, nasal bleeding, anemia, and oedema [10–21]. In turn, intrinsic characteristics of the individual, such as age and HIV co-infection, in addition to other comorbidities, the relapsing course of the disease [22,23] and high parasite load [23,24], although useful in understanding the VL case-fatality phenomenon, are insufficient markers to guide an effective and preventive intervention of the fatal outcome.

In parallel, the current urban pattern of the disease represents a difficult and continuous challenge for VL control in Brazil. Leishmaniasis has strong links with poverty, due to poor housing conditions and deteriorated environmental sanitation, and with low income, gender imbalance, displacements, immunosuppression, and poor nutrition, among other determinants [25]. Traditionally, this link with poverty has been explained by the risk of acquiring the infection provided by the promiscuous proximity between vector reservoir and host, and also by the increased risk of illness development in individuals immunocompromised by malnutrition, comorbidities and co-infections [26]. However, other aspects need to be added to this analysis, such as the association between poverty and lack of access to timely diagnosis and treatment. Thus, the main hypothesis that guides this analysis is the influence of factors related to the socio-economic contextual conditions and access to the health system on the VL case-fatality. To achieve this goal, this study aims to investigate the association between factors at the both FU and individual levels and VL case-fatality, after analysing the temporal trend and the spatial dependency for VL case-fatality. The understanding of this macro determinants can guide public policies directed at decreasing VL case-fatality.

## Methods

It is a population-based, ecological study addressing the temporal trend and spatial distribution of case-fatality for VL cases notified in Brazil from 2007 to 2017, followed by a multilevel analysis including individual and contextual factors related to the VL case-fatality.

### Ethics statement

The study was approved by the Ethical Review Board of the Instituto René Rachou, Fundação Oswaldo Cruz—Fiocruz (Approval number 3.331.474, CAAE 12048219.8.0000.5091) with exemption from individual consent as it is a secondary database analyzed after anonymization of personal information.

## Source of data

The database sources are the aggregated and individual dataset of VL cases reported to the Brazilian health epidemiological surveillance system (DATASUS) http://tabnet.datasus.gov.br/cgi/deftohtm.exe?sim/cnv/obt10uf.def), the socioeconomic and demographic indicators provided by the Brazilian Institute of Geography and Statistics (IBGE) http://tabnet.datasus.gov.br/cgi/deftohtm.exe?ibge/cnv/popuf.def) after the last Brazilian population census (2010), besides the basic health data available in two Brazilian health information systems, Indicators and Basic Data http://www2.datasus.gov.br/DATASUS/index.php?area=02 (IDB 2012) and the National Register of Health Establishments—CNES http://www2.datasus.gov.br/DATASUS/index.php?area=0204, accessed in August, 2019.

In Brazil, VL is a disease of compulsory notification, i. e., in the case of clinically suspected VL, health professionals must fill in a specific case notification form in the Reportable Disease Information System (SINAN). Initially, demographic and clinical manifestations (signs and symptoms) are added to the dataset. Later, additional information such as laboratorial exam results, date of beginning of treatment, initial drug used for treatment, drug used following failure of the initial therapy and outcome (cure or death from VL) are provided by epidemiological surveillance professionals in the municipalities. However, details on treatment-related toxicity, early interruption and sequential treatments are not included in the notification form, which prevents a deeper analysis of their influence on the evolution of cases. The individual variables extracted from the VL case notification form make up the variables at the individual level. Additionally, public domain epidemiological, socioeconomic and health system access indicators set for each administrative political unit in Brazil, the Federated Unit (FU), were also considered in the analysis. Individual variables and the demographic, socioeconomic, health access and epidemiological indicators explored in this analysis are presented in Tables 1 and 2.

## Population eligibility applied to the Brazilian VL dataset

The inclusion criteria in this study were: (i) the VL case notification for SINAN was performed between 2007 and 2017; (ii) the VL case is a person resident in Brazil; (iii) the VL diagnosis was considered confirmed; (vi) the outcome was known (death from VL or not was informed in the SINAN dataset).

## Analysis strategy and statistical methods

The classical regression models presuppose independence among observations, which cannot be assumed analyzing information collected over time in individuals within spatial units such as FUs. Thus, the temporal and spatial analyzes were done a priori to verify these dependencies in order to define the best strategy of incorporating these terms in the subsequent analyzes.

**Temporal analysis.**   The VL case-fatality dependence on time was evaluated considering the year of notification the unit of analysis, from 2007 to 2017. A smoothing function was used in the year variable, through a generalized additive model (GAM), to verify the dependence among observations over time, to recognize the functional behavior of the "year" variable and to assist in building the model including both individual and contextual data. In this model, the response variable was the number of deaths in year i with Poisson distribution, offset term the number of confirmed VL cases in year i, and a spline function in the continuous time variable [27,28].

**Spatial analysis.**   To assess dependence on space, the Brazilian FUs were taken as units of analysis. In this analysis, the case-fatality was estimated for each FU as the number of deaths caused by VL in the period, divided by the number of confirmed VL cases, in that FU,

**Table 1. Potential risk factors for case-fatality from VL explored at individual level (variables extracted from the VL case notification form).**

| Variable | Description |
|---|---|
| **Variables generated from database** | |
| Age | Difference between notification date and date of birth |
| Time between symptom onset and notification | Difference between dates of the symptom onset and notification date. For records where the date of the symptom onset was equal to the date of notification or birth, the field was considered missing |
| Time between diagnosis and initiation of treatment | Difference between dates of diagnosis and start of treatment. For records where the date of diagnosis was equal to the date of birth, the field was considered missing |
| Time between treatment initiation and death | Difference between treatment start date and date of death. For records where the date of treatment initiation was later to the date of death, the field was considered missing |
| **Demographic and clinical information present in the notification form** | |
| Gender | (Female, male) |
| Race according to Brazilian Institute of Geography and Statistics (IBGE) | (White, black, yellow, parda, indigenous) |
| Schooling | Education level |
| Local of residence | Rural or urban |
| VL case classification | New VL case (primary), relapsing case, transfer |
| Diagnostic criterion | Laboratory confirmed, clinical-epidemiological criteria |
| HIV co-infection | Yes, no |
| Clinical manifestations | Fever, weakness, splenomegaly, hepatomegaly, weight loss, bleeding, cough, jaundice (the presence or absence and the total sum of symptoms and signals were analyzed) |
| Parasitological confirmation of VL | Leishmania presence in a direct exam or culture obtained from tissue, blood or bone marrow aspirate |
| Indirect immunofluorescence test (IFAT) | Positive, negative |

multiplied by 100. An ecological regression model or spatial model was used under a Bayesian approach through hierarchical models, using the conditional intrinsic Gaussian autoregressive model (CAR model) [29–31]. In this model we evaluate the contribution of two random terms to each FU, a spatially structured–uf.nu term (considering spatial correlation between the FUs) and a non-spatially structured–uf.theta term, which considers the influence of possible factors at the FU level that have not been measured or observed–random intercept [32]. The number of deaths from VL followed a Poisson probability distribution. In order to consider the different number of confirmed VL cases in each FU and its different age distributions, and to estimate the VL case-fatality, the term offset (the number of VL cases which would be observed if age distribution were equal to the standard population) was added to the modeling, after standardization by direct method, based on the Brazilian population in 2010. The calculated number of VL cases that would be observed if age distribution were equal to the standard population, at each FU, were used in the term offset. The inference on the parameters and hyperparameters were approximated by the deterministic procedure Integrated Nested Laplace Approximations (INLA) [33]. In sequence, the appropriateness was verified by analyzing the significance of the hyperparameters, through the posterior probability distribution, besides comparing the number of deaths observed with the number of deaths estimated by the model through the map.

**Multilevel analysis.** Considering the main hypothesis to be investigated in this study, that variables related to the context of the Brazilian FUs can influence the risk of dying among VL cases, in addition to the individual factors, fixed and random effects of the factors of interest were assessed using mixed or multilevel effects models (GLMM).

**Table 2. Potential contextual risk factors for VL case-fatality explored at FU level (demographic, socioeconomic, health access and epidemiological indicators set for each Brazilian FU).**

| Variable | Description |
|---|---|
| **Socioeconomic and demographic variables** | |
| Gender proportion | Number of men for each group of 100 women in the total FU population |
| Race proportion | Percentage of self-reported non-white people in total FU population |
| Proportion of elderly | Percentage of persons aged 60 years or over in total FU population |
| Proportion of children under 5 years | Percentage of persons aged 5 years or less, in total FU population |
| Illiteracy rate | Percentage of persons aged 15 or older, who cannot read and write at least one single ticket in the mother language in the total of the resident FU population, in the same age group |
| Urbanization level | Proportion of total FU population living in urban areas |
| Population growth rate | Percentage of annual average increase in population living in a given FU, during the period considered. |
| Life expectancy at birth | Average number of years of life expected for a newborn, maintaining the mortality pattern |
| Child mortality | Number of deaths of children under one year of age per thousand live births in the FU in the year considered |
| Average household income per capita | Average household incomes per capita. Sum of monthly household income, in Real, divided by number of house's residents in the FU |
| Water supply network | Percentage of resident FU population served by water supply network, with or without home plumbing |
| Access to sewage | Percentage of resident FU population with access to household connection to sewage system or septic tank |
| Garbage collection | Proportion of the FU population served with garbage collection |
| GINI index | Concentration index for household income distribution (per capita) |
| PIB per capita | Gross domestic product divided by the FU inhabitants' number |
| HDI | Human development Index (2010) |
| **Health care access variables** | |
| Number of medical doctors | Number of active medical doctors, per thousand FU inhabitants |
| Health units | Number of health units in FU |
| Total number of hospital beds | Total number of hospital beds per FU |
| Number of emergency beds | Number of emergency beds per thousand inhabitants in FU |
| Health expenditure | Expenditure on actions and public health services per inhabitant |
| Family Health Program (FHP) | Number of multidisciplinary teams working at the Family Health Program |
| **Epidemiological variable** | |
| VL standardized incidence | Age-standardized VL incidence rate |

VL: visceral leishmaniasis FU: (Brazilian) federated unit

Based on both the result of CAR model, where there was a significant unstructured term (random intercept), and on the result of GAM model, where there was a linear effect of time, all adjusted mixed models built, univariate and multiple, considered these two terms. The mixed modeling was built assuming that the distribution of the outcome variable death by VL had the binomial probability distribution and using the logit link function. As an exploratory analysis prior to the modeling process, the variables of interest were described as proportions, median and 25–75% interquartile range (IQR25-75%). The associations between each variable with outcome (death from VL) and among the explanatory variables—as a previous strategy for detecting possible multicollinearity in multiple models, were explored by parametric or

non-parametric hypothesis tests (tests for proportions—chi-square test; test of means or medians—t-test or Wilcoxon, analysis of variance or Kruskal-Wallis, and correlation coefficients—Pearson and Spearman, depending on the data's distribution pattern). When a high correlation between two explanatory variables (> 85%) was detected, those with the lowest p value in the univariate model were selected for the multiple models [34].

As the first step in the modeling process, a null model was adjusted (only with random intercept). Based on this model, the proportion of variance in VL case-fatality due to intra-FU correlation was estimated. In addition to it, a linear time term was inserted as a fixed effect and it was taken as the basic model. In the next step, the variables measured at the FU's level and, subsequently, those at the individual level were tested individually, being added to that basic model (random intercept + temporal term + variable).

In the univariate models, the association between each variable and outcome (death from VL) was verified. Simultaneously, for the continuous explanatory variables, the functional form of this relationship was assessed using smoothing functions (thin plate regression spline) through generalized additive mixed modeling–GAMM. Based on GAMM, a graphical (visual) analysis was performed in order to allow the identification of an association between an explanatory and an outcome variable (binomial), in addition to inferring in which ranges of values this association is significant, which can be declared when ranges of values of the curve do not belong to the estimated confidence interval and contain included zero value. If the association is significant, we can also infer whether it is positive or negative. This strategy was used to make categorizations, or to adjust univariate linear models, for intervals in which the functional form was linear, using a subset of the database. In univariate analysis, variables at the individual and at the FU level were considered candidates to the multiple modeling at the 20% significance level. Finally, the inclusion of variables whose effects were linear or categorized was tested, with fixed coefficients. Thus, the significant variables were introduced into the basic model using a manual step-by-step forward selection procedure in decreasing order of significance. The exclusions due collinearity were made pair by pair, in case of two related variables are together in the same model. All excluded variables were retested, one by one, in the final model. The contribution of each added variable was tested by testing the likelihood ratio between nested models (when one model contains all the terms of the other, and at least one additional term). After that, interactions between the significant predictor terms were tested. Finally, in order to explain part of the variability among the FUs, random coefficients terms were tested to assess the FUs impact on the contextual variables that remained in the multiple analysis model, the same way as those that did not enter in the multiple model, as they could become significant in the presence of other variables. The individual significance of each variable in the best fitted model was considered at 5%, and *odds ratio* and 95% confidence interval were estimated. All the analyses were performed using the statistical software R (packages mgcv, INLA, lme4, ggplot2) [35] and the Statistical Package for the Social Sciences (SPSS), version 24. The maps were built using the free and open source R software (https://www.R-project.org/.) based on shapefiles obtained from Instituto Brasileiro de Geografia e Estatística-IBGE) (https://portaldemapas.ibge.gov.br/portal.php#homepage).

## Results

From January 1, 2007 to December 31, 2017, 102220 suspected cases of VL were reported in Brazil in 27 Federated Units. Of this total, 41,204 (40.3%) cases were confirmed. The national average VL incidence in the 11 year-period was 1.9 cases per 100,000 inhabitants, ranging from 1.69 to 2.16/100,000 inhabitants. Of the confirmed cases, 32723 (79%) have a clinical evolution registered in SINAN, either "cure" or "death", the latter classified as "from VL" or "from

**Table 3. Number of visceral leishmaniasis cases, deaths, incidence and case-fatality, Brazil, 2007–2017.**

| Variables | YEAR OF NOTIFICATION | | | | | | | | | | | Total |
|---|---|---|---|---|---|---|---|---|---|---|---|---|
| | 2007 | 2008 | 2009 | 2010 | 2011 | 2012 | 2013 | 2014 | 2015 | 2016 | 2017 | |
| VL incidence (by 100,000 inhabitants) | 1.89 | 2.11 | 2.04 | 1.95 | 2.14 | 1.69 | 1.74 | 1.86 | 1.76 | 1.69 | 2.16 | 1.91 |
| VL confirmed cases | 3565 | 3991 | 3894 | 3704 | 4107 | 3269 | 3472 | 3733 | 3558 | 3455 | 4456 | 41204 |
| Clinical outcome known | 2846 | 3093 | 3000 | 3031 | 3469 | 2639 | 2706 | 2779 | 2800 | 2754 | 3606 | 32723 |
| Death from VL | 192 | 221 | 236 | 233 | 266 | 217 | 232 | 248 | 280 | 272 | 338 | 2735 |
| Case-fatality (%) | 6.75 | 7.15 | 7.87 | 7.69 | 7.67 | 8.22 | 8.57 | 8.92 | 10.00 | 9.88 | 9.37 | 8.36 |

VL: visceral leishmaniasis

other causes". The annual distribution of confirmed cases, number of deaths, VL incidence and case-fatality is presented in Table 3 and Fig 1a. On average, 2974 cases/year were confirmed, with a standard deviation (SD) of 313 cases. There was the average of 249 ± 39 death/year and the VL case-fatality ranged from 6.7 to 10.0%, with an average of 8.4% (± 1.1). The highest levels of case-fatality were observed in 2015 (10.0%) followed by 2016 (9.9%).

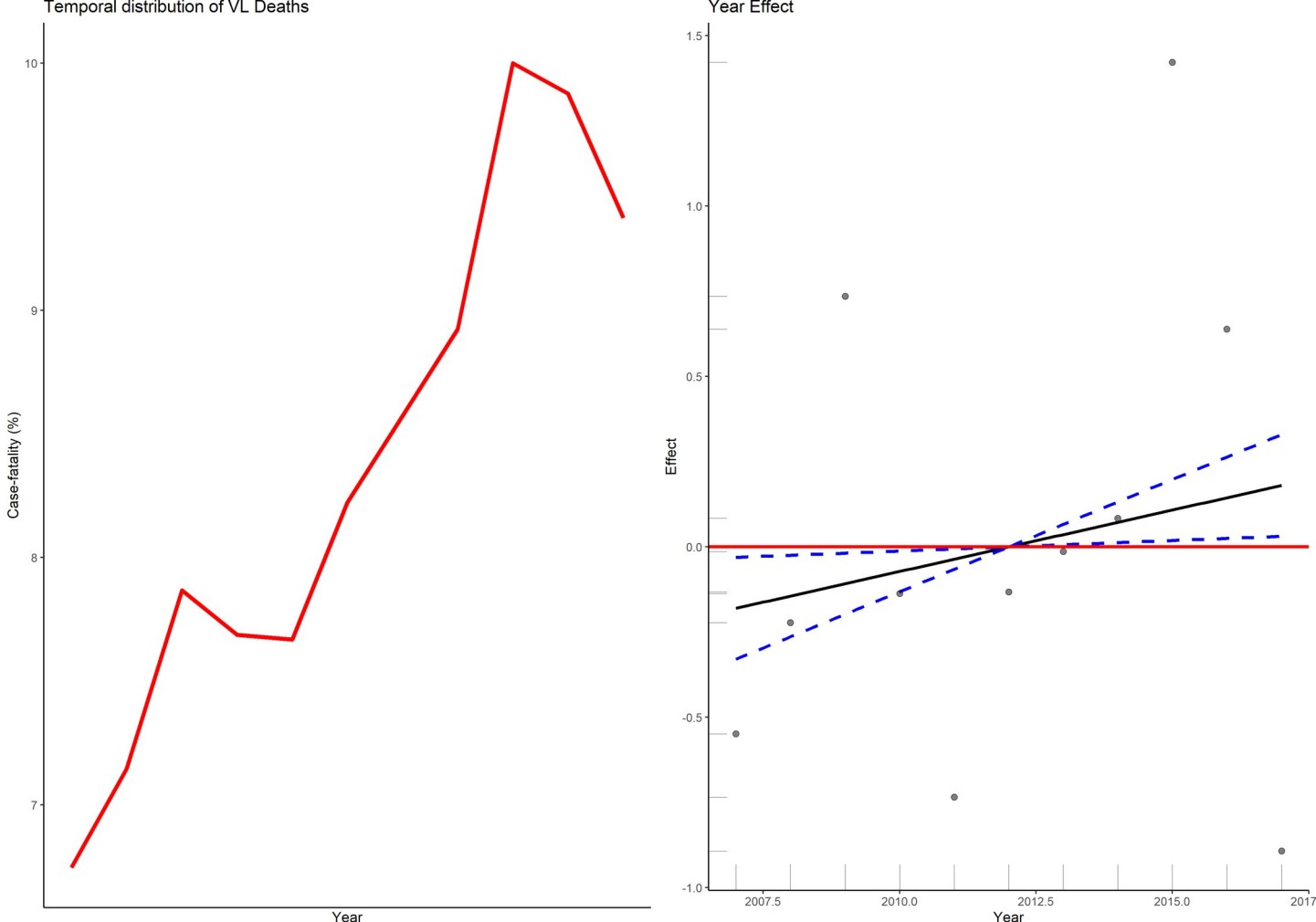

**Fig 1.** Observed annual distribution of case-fatality rate (a), and temporal effect estimated by generalized additive model (b).

### Temporal analysis

The temporal effect estimated by the generalized additive model is shown in Fig 1. In Brazil between 2007 and 2017, the smoothing function of the GAM model showed a linear and increasing tendency for case-fatality over time. That linear effect was estimated through a Generalized Linear Model, where $e^{0.036059} = 1.04$, or an annual average case-fatality increases of 4% (CI95%: 2.5–4.9%). This linear effect of time was added to the mixed model later.

### Spatial analysis

Fig 2 (left) shows the distribution of deaths observed among the FUs, while Fig 2 (right) shows the spatial distribution of case-fatality among them. The distribution of case-fatality is heterogeneous in Brazil, with higher rates in the north and south, in FUs with the lowest VL incidence (Fig 2 and S1 and S2 Tables). The spatial effect of VL case-fatality was evaluated based on the spatial model (CAR) analysis, a significant unstructured effect or random intercept—uf. theta (Fig 3B) was observed, because the zero value for variance of this term (hyperparameter) has a low probability of occurrence (small amount of data under the curve) in its a posteriori distribution, differently from the spatially structured term—uf.nu (Fig 3A). This finding reveals that although there is no similarity between neighboring FUs in terms of case-fatality, the presence of contextual variables at the level of each FU influencing VL case-fatality rate but not directly observed (random intercept effect of the FU). The model presented a good data-fit considering the significant variability captured and the similarity of deaths observed and predicted (Fig 3c and 3d). Based on this result, we included FU-level random intercepts in our multi-level models of VL case-fatality described in the following section.

### Multilevel analysis

The analysis of the individual notification dataset consisting of the total of 32723 VL cases reveals a predominance of men (63.8%) and a median age of 16 years (IQR25-75% 2–39 years). The HIV co-infection rate was 7.7%, an information available in 75% of the notified

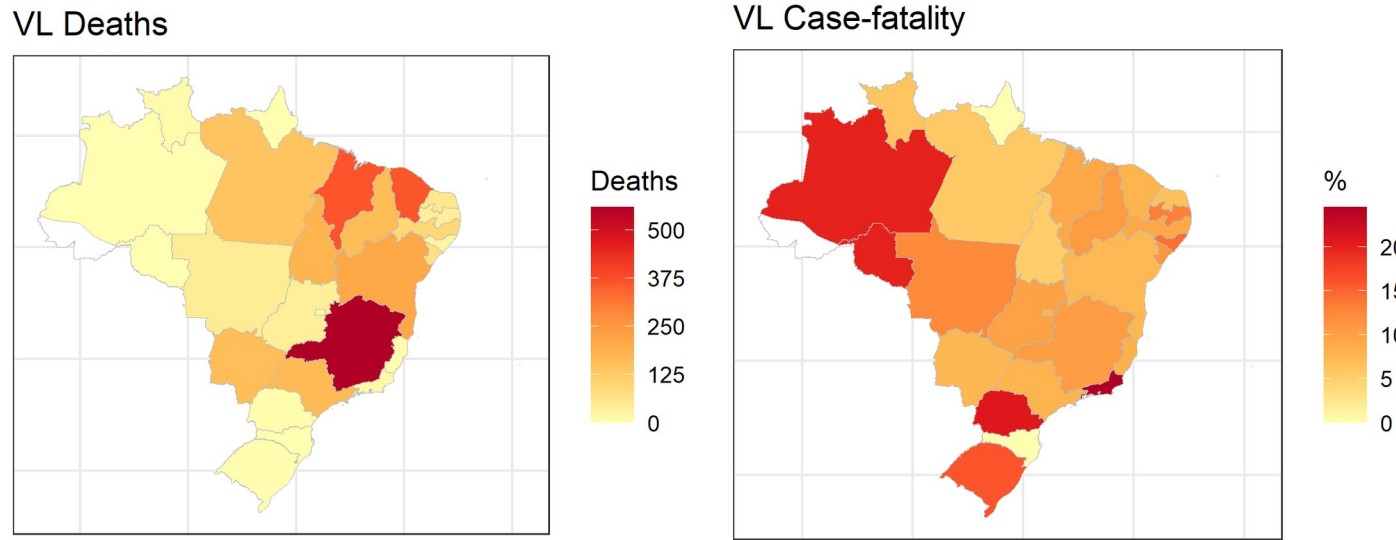

**Fig 2.** Spatial distribution VL deaths (left) and case-fatality in the FUs (right), Brazil, 2007–2017. The maps were built using the free and open source R software (https://www.R-project.org/.) based on shapefiles obtained from Instituto Brasileiro de Geografia e Estatística-IBGE) (https://portaldemapas.ibge.gov.br/portal.php#homepage).

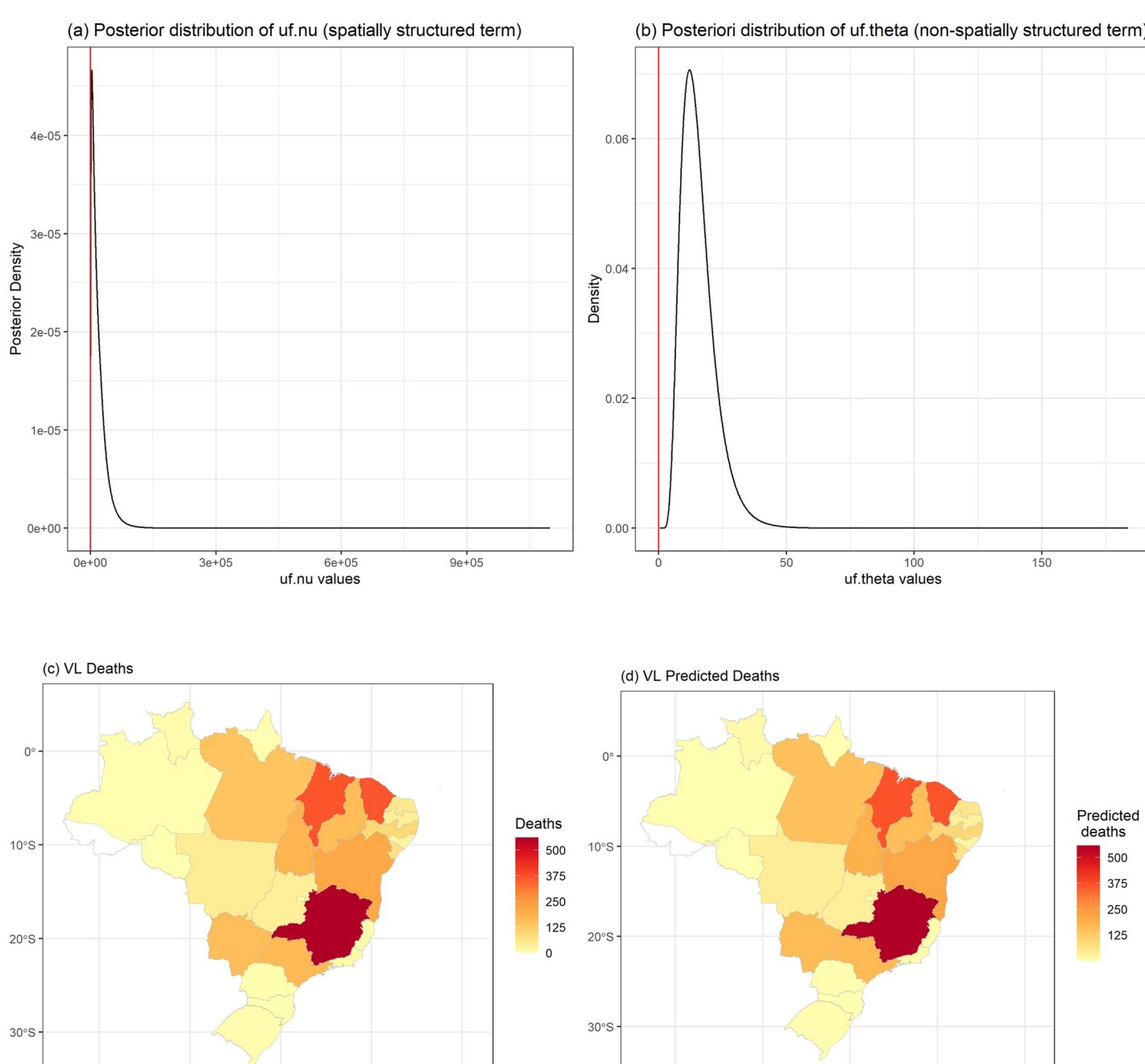

**Fig 3.** Posteriori distributions of the structured spatial term–uf.nu (a) and the unstructured spatial term—uf.theta (b) hyperparameters. Observed deaths (c) and predicted deaths (d) fitted by CAR model, Brazil, 2007–2017. The maps were built using the free and open source R software (https://www.R-project.org/.) based on shapefiles obtained from Instituto Brasileiro de Geografia e Estatística-IBGE) (https://portaldemapas.ibge.gov.br/portal.php#homepage).

cases (25% of missing data). After the exclusion of cases reported as death from other causes, VL cases with outcome defined as cure or death from VL totaled 31856 cases. In the 11-year study period, 2735 patients died from VL (case-fatality of 8.4%).

From a clinical perspective, several differences were observed between fatal and non-fatal VL cases, significance defined by a p value of 0.05. The medians of age and the interval from

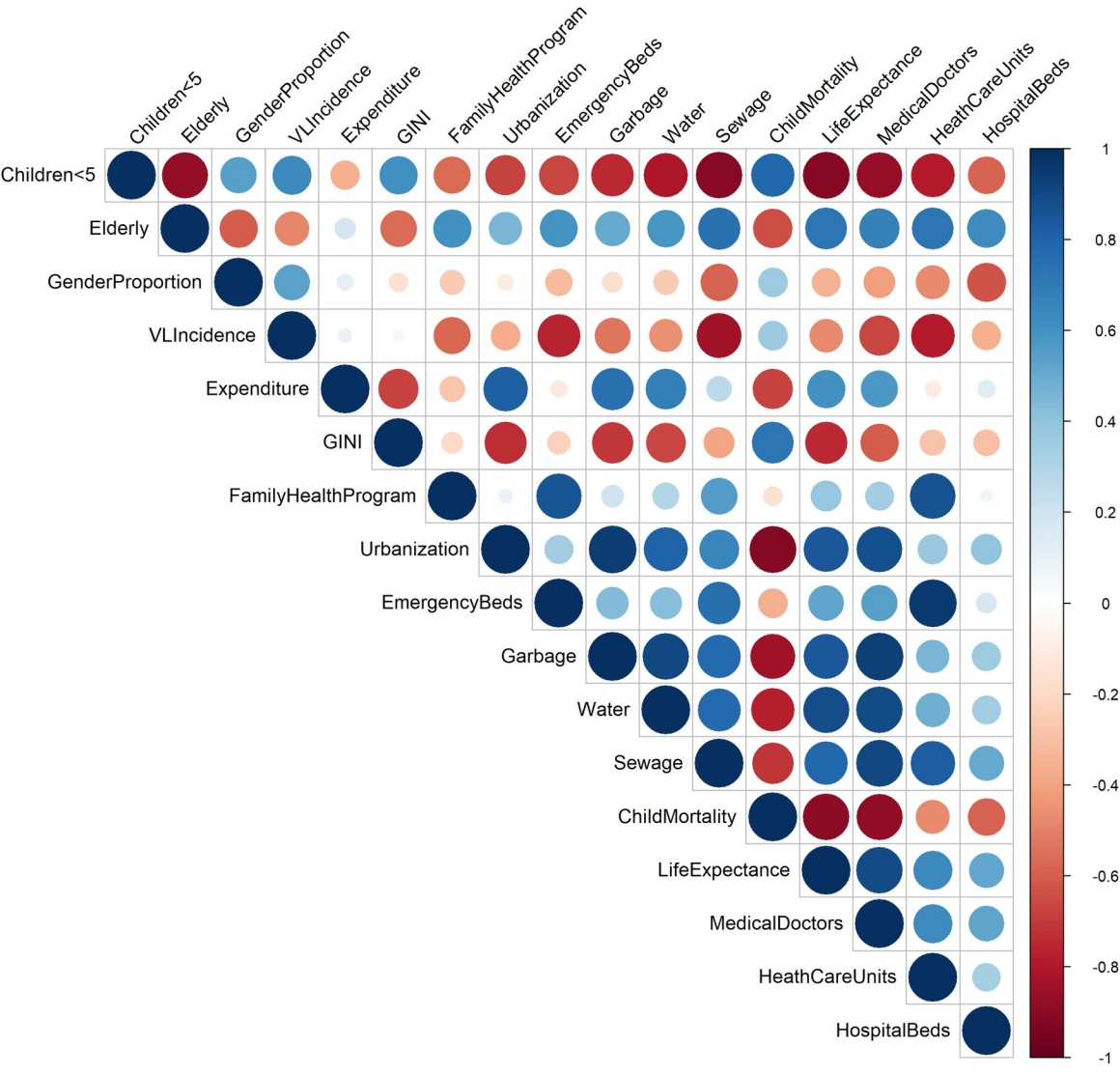

**Fig 4. Correlation matrix for contextual variables.**

the onset of symptoms to VL treatment among patients with fatal evolution were 39 (IQR25-75% 7–58) years and 33 (IQR25-75% 16–76) days, significantly different from those observed for VL cured group: 13 (IQR25-75% 2–36) years and 30 (IQR25-75% 15–59) days, respectively. Concerning to the diagnostic approach, fatal VL cases had a higher percentage of diagnostic confirmation by parasitological test (36% versus 31%, p<0.001). It was also observed a trend in favor of a lower rate of IFAT serological testing (49% versus 51%, p = 0.07) among fatal VL cases and no difference for other serological tests use between the two groups (43% for both fatal and cured VL cases).

Multicollinearity analysis were performed a priori to find associations between explanatory variables. Among the variables at the individual level, time between the first symptom and the VL notification and time from VL notification to treatment were highly correlated (r = 0.985). In the same way, there was high correlation among several contextual variables (Fig 4), such as water access and garbage collection, correlated to many other variables. Variables with

Spearman correlation coefficients above 85% were not included together in the multiple models, only those with greater associations with the outcome VL case-fatality.

The standard error estimation in the null model (random intercept) was 0.31 for the random effect at the level of FUs and 1.00 for the residuals (result not shown), resulting in an intra-FU VL case-fatality correlation of 23.7% (0.31 / (1.00 + 0.31)). The presence of this correlation confirms the need to include FU conglomeration in the multilevel model. This intra-FU correlation estimates the proportion of variance of the outcome that occurs within the FU's compared to the total dataset variability.

The mixed univariate models were used to assess the significance of each variable independently, in the presence of the temporal term and the random intercept at the FU level.

We simultaneously evaluated the functional form of continuous variables, using smoothing functions in univariate GAMM models. In Fig 5, red lines represent a null effect, meaning that if the line crosses the confidence interval, the effect is null in that region. Green lines represent the setpoints defined for the original variable. Fig 5a shows that the variable time from the first VL symptom to notification had a positive and approximately linear association with VL case-fatality up to 200 days, thus, the variable was categorized at this point and the effect on the VL outcome estimated in Table 4. Assuming the estimated effect as the effect of one variable in addition to the spatial random intercept and temporal term, VL cases notified after 200 days of the onset of symptoms have a chance of death 1,63 times the chance of those VL cases notified before 200 days of symptoms. The GAMM model indicated a risk related to age over 20 years and this variable was dichotomized (<20 years; > = 20 years) (Fig 5b). The estimated crude effect shows a risk of patients over 20 years three times the risk of death of patients under 20 years (Table 4). Race, local of residence (rural or urban) and VL case classification (primary or relapse) were not associated with outcome. The time between VL symptoms onset and treatment was significantly related to case-fatality up to 500 days (Fig 5c), above this value, the confidence interval was very wide, leading to inaccuracy in the estimates, thus, the effect was estimated up to this setpoint (Table 4) and values above this point were discarded. Considering a non-linear association between standardized VL incidence and VL case-fatality (Fig 5d), this variable was categorized as following: up to 60, 60 to 160, and above 160 cases per 100,000 inhabitants. Concerning to VL incidence, VL cases from FU with less than 60 VL cases/100,000 inhabitants had more risk of fatal outcome compared to VL cases from FU with more than 160 cases/100,000 inhabitants.

On the other hand, the number of emergency beds/100,000 inhabitants was significantly associated to VL case-fatality up to 1,000 beds/100,000 inhabitants (Fig 5e). The effect was estimated up to this setpoint (Table 4) and an inverse association was confirmed: the greater the number of emergency beds, the lower the VL case-fatality (Fig 5e). Inversely, the absolute number of hospital beds exhibited a linear and positive association with VL case-fatality (not shown). The number of multi-professional teams working at Family Health Program (FHP) was inversely related to the risk of dying from VL, for the interval of 200 to 600 FHP teams/FU. Thus, this variable was evaluated up to the limit of 600 FHP teams/FU, dichotomized at < = 200 and> 200 FHP teams/FU (Fig 5f). Access to sewage, a variable that was truncated to 29%, exhibited a positive effect of 1.23 (1.14–1.33) on VL case-fatality. In Fig 5h, which represents the variable garbage collection, the analysis was made for values greater than 93% and the effect was protective on VL case-fatality was 0.54 (0.36–0.81). For the number of medical doctors, a progressive effect up to 1.5 per 1,000 inhabitants could be observed, stabilized afterwards (Fig 5i). This variable has been dichotomized and a protective effect against VL case-fatality was confirmed for more than 1.5 medical doctors/1,000 inhabitants. Concerning to the health expenditure, the GAMM models revealed three ranges of values with different associations with the outcome, which guided discretization (Fig 5j). However, no significant

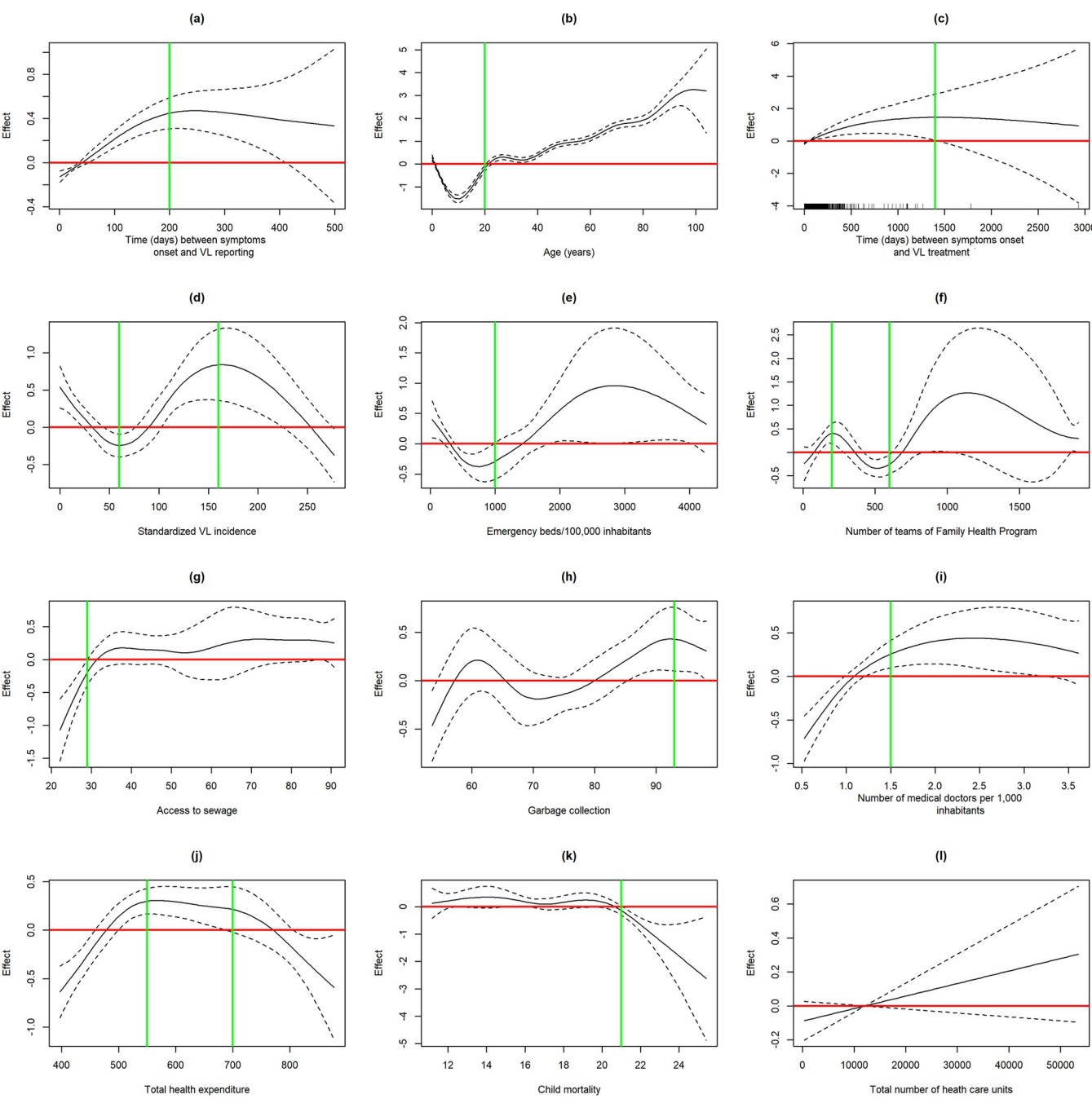

**Fig 5. Effect of continuous variables on VL case-fatality (GAMM models).**

association could be confirmed in univariate analysis (Table 4). Variables of the epidemiological context were also explored. Child mortality, analyzed for mortality rates higher 21%, had an apparent inverse effect on VL case-fatality (Fig 5k), but it was no demonstrated in the simple mixed model (Table 4). The proportion of elderly people, life expectancy and degree of urbanization were linear and directly associated to VL case-fatality. In turn, sex ratio, the proportion of children under 5 years and GINI index had an inverse linear effect on VL case-fatality. Despite a linear association with VL case-fatality (Fig 5l), no association could be

**Table 4. Association between contextual/individual factors and VL case-fatality in univariate analysis (Generalized linear mixed models).**

| | Cure (%) | Death (%) | Univariate analysis |
|---|---|---|---|
| | n = 29221 | n = 2735 | OR (95% CI) |
| Number of VL cases/year, mean (SD) | 3140 (271) | 249 (37.5) | 1.038 1.025–1.051) |
| Age | | | |
| < 20 years (base) | 16506 (51.8) | 820 (2.6) | - |
| > = 20 years | 12600 (39.6) | 1911 (6.0) | 3.01 (2.76–3.28) |
| Gender | | | |
| Female (base) | 10656 (33.5) | 926 (2.9) | - |
| Male | 18450 (58.0) | 1805 (5.7) | 1.10 (1.01–1.20) |
| Race | | | |
| White (base) | 4632 (15.8) | 430 (1.5) | 0.98 (0.87–1.10) |
| Non-white | 22171 (75.7) | 2032 (6.9) | |
| Local of residence | | | |
| Urban/peri-urban (base) | 22092 (71.3) | 2063 (6.7) | 0.98 (0.89–1.09) |
| Rural | 6244 (20.1) | 569 (1.8) | |
| VL case classification | | | |
| Primary (base) | 26744 (86.3) | 2481 (8.0) | 0.94 (0.79–1.12) |
| Relapsing/transfer | 1599 (5.2) | 150 (0.5) | |
| Bacterial co-infection | | | |
| No (base) | 20545 (70.7) | 1398 (4.8) | - |
| Yes | 6050 (20.8) | 1078 (3.7) | 2.51 (2.31–2.73) |
| HIV co-infection | | | |
| No (base) | 19971 (83.3) | 1719 (7.2) | - |
| Yes | 1999 (8.3) | 279 (1.2) | 1.52 (1.33–1.74) |
| Schooling | | | |
| Up to elementary school (base) | 668 (7.0) | 164 (1.7) | - |
| Above elementary level | 7868 (82.7) | 812 (8.5) | 0.44 (0.36–0.53) |
| Number of VL symptoms (%) | | | |
| Up to 3 symptoms (base) | 6986 (21.9) | 402 (1.2) | - |
| 4–6 symptoms | 18015 (56.5) | 1440 (4.5) | 1.41 (1.25–1.58) |
| = /> 7 symptoms | 4105 (12.8) | 889 (2.7) | 3.89 (3.43–4.41) |
| Interval between symptoms onset and VL reporting | | | |
| = /> 200 days (base) | 816 (2.7) | 123 (0.4) | - |
| < 200 days | 26460 (88.7) | 2423 (8.1) | 0.64 (052–0,77) |
| Interval between symptoms onset and VL treatment (limited to 500 days), median days (IQR 25–75%) #1 | 30 (15–58) | 33 (16–73.3) | 1.003 (1.001–1.004) |
| Parasitological confirmation (%) | | | |
| Negative (base) | 2606 (8.3) | 203 (0.64) | - |
| Positive | 9343 (29.7) | 982 (3.1) | 1.37 (1.17–1.61) |
| Not performed | 16819 (53.5) | 1490 (4.7) | 1,19 (1.02–1.39) |
| Proportion of children under 5 years, % (IQR 25–75%)* | 7.6 (7.5–8.9) | 7.6 (6.5–8.2) | 0.73 (0.66–0.81) |
| Proportion of elderly, % (IQR 25–75%)* | 10.4 (8.6–10.8) | 10.6 (8.6–11.6) | 1.16 (1.09–1.24) |
| Gender proportion (IQR 25–75%)* § | 96.9 (95.1–99.3) | 96.9 (95.1–98.4) | 0.95 (0.92–0.98) |
| Standardized VL incidence (VL cases/100,000 inhabitants)* | | | |
| <60 (base) | 14990 (47.1) | 1502 (4.7) | - |
| 60–160 | 10788 (33.9) | 1043 (3.3) | 0.89 (0.68–1.18) |
| >160 | 3328 (10.4) | 186 (0.6) | 0.61 (0.39–0.94) |
| Access to garbage collection (limited to access above 93%) (IQR 25–75%) #2 | 98.2 (98.2–98.2) | 98.2 (98.2–98.2) | 0.54 (0.36–0.81) |

*(Continued)*

**Table 4.** (Continued)

| | Cure (%) | Death (%) | Univariate analysis |
|---|---|---|---|
| | n = 29221 | n = 2735 | OR (95% CI) |
| Water supply access* % (IQR 25–75%) | 78.4 (74.2–85.7) | 78.3 (74.2–86.7) | 1.02 (1.01–1.03) |
| Access to sewage (limited to 29%) (IQR 25–75%) #3 | 28.2 (25.1–28.2) | 28.2 (25.1–28.7) | 1.23 (1.14–1.33) |
| Health expenditure (reais per capita)* | | | |
| <550 (base) | 14289 (44.9) | 1315 (4.1) | - |
| 550–700 | 7547 (23.7) | 874 (2.7) | 1.14 (0.89–1.47) |
| > 700 | 7270 (22.8) | 542 (1.7) | 0.95 (0.71–1.28) |
| Number of medical doctors/1000 inhabitants* | | | |
| = < 1.5 (base) | 6950 (21.8) | 792 (2.5) | - |
| >1.5 | 22156 (69.6) | 1939 (6.1) | 0.74 (0.56–0.97) |
| Hospital beds/FU*, median (IQR 25–75%) | 2.2 (2.1–2.3) | 2.2 (2.1–2.3) | 2.68 (1.93–3.71) |
| Number of FHP teams/FU* #4 | | | |
| < = 200 (base) | 9481 (43.2) | 871 (3.9) | - |
| 200–600 | 10663 (48.6) | 917 (4.2) | 0.67 (0.54–0.82) |
| Health units/FU*, median (IQR 25–75%) | 5988 (3243–12802) | 6516 (3243–28244) | 1.00 (0.09–1.00) |
| Urbanization level*, % (IQR 25–75%) | 75.4 (72.1–85.3) | 76.6 (72.1–85.3) | 1.02 (1.01–1.03) |
| Child mortality over 21%* (IQR 25–75%) | 17.2 (16.2–21.0) | 17.0 (16.2–20.7) | 0.41 (0.08–2.22) |
| Life expectance at birth*, years (IQR 25–75%) | 71.9 (69.9–74.1) | 72.4 (69.9–75.5) | 1.06 (1.01–1.12) |
| GINI∞ index* (IQR 25–75%) | 0.62 (0.58–0.63) | 0.61 (0.56–0.63) | 0.01 (0.00–0.21) |

VL: visceral leishmaniasis FU: (Brazilian) federated unit OR: odds ratio CI95% 95% confidence interval IQR25-75%: 25–75% interquartile range SD: stand deviation § number of men for 100 women in the total FU population ∞ concentration index for household income distribution per capita * Indicator refers to the FU where VL cases were reported FHP: Family Health Program #1 included VL cases with time between symptoms onset and VL treatment lower than 500 days, above this value, the confidence interval was very wide, leading to inaccuracy in the estimates #2 included VL cases reported by FU with garbage collection rate higher 86% #3 included VL cases reported by FU with access to sewage up to 29% #4 included VL cases reported by FU with FHP teams up to 600.

confirmed between the total number of heath care units and VL outcome. The average household income and the gross domestic product, both per capita, in addition to HDI, sewage and water access, illiteracy and race proportions, population grow rate were not associated with VL case-fatality. Variables highly related to others of the same group (demographic or health care access) and not associated to outcome were not shown in the Table 4.

Considering the variation of the effect of variables at the individual level on VL case-fatality among FUs, the significance of random slopes for these variables was tested but no association was identified. The variables that remained significant simultaneously at the 5% level, resulting in the best fitted model, are presented in Table 5, where the adjusted OR and 95% confidence intervals are shown. Significant interaction between sex and age variables have confirmed a high risk of the stratum men over 20 years, in which the risk of death increases 1.744 times.

Fig 6A shows the estimated error for the random effect of the intercept in the basic model (without covariables). FUs with above-expected case-fatality (Piauí/PI and Minas Gerais/MG) and FUs with below-expected case-fatality (Tocantins/TO, Pará/PA, Rio Grande do Norte/RN, Bahia/BA, Ceará/CE) were identified. For all FU's excepted RN, the difference in case-fatality is explained by the variables that remained in the final multiple model. For RN, even after the model adjustment by covariates, the VL case-fatality remained lower than expected (Fig 6B), which may be related to local conditions on a spatial scale lower than that used in this analysis, therefore not measured.

**Table 5. Final multiple modeling for factors related to VL case-fatality.**

| Variable | OR (CI 95%) |
| --- | --- |
| Year of VL reporting | 1.034 (1.010–1.059) |
| Interval between symptoms onset and VL reporting < 200 days | 0.745 (0.554–1.059) |
| Age = /> 20 years | 3.471 (2.403–5.014) |
| Number of VL symptoms 4–6 | 1.764 (1.343 2.316) |
| Number of VL symptoms = />7 | 4.199 (3.135–5.623) |
| Schooling above elementary level | 0.561 (0.455–0.692) |
| Male gender | 0.425 (0.257–0.702) |
| Bacterial co-infection | 1.870 (1.595–2.193) |
| Age above 20 years and male gender | 1.744 (1.025–2.970) |

## Discussion

The main observation of this study was the confirmation of a linear increase in the VL case-fatality in Brazil despite several attempts to improve the VL diagnostic and therapeutic approach in recent years. The lack of the expected impact after incorporating of rapid tests and liposomal amphotericin B, requires two reflections: the first, on the real spreading of these interventions in a country of continental extension and, the second, on the role of delay in diagnosis and the toxicity caused by antimony derivatives in the VL case-fatality in Brazil. In addition to these two points, a third factor must be considered to understand the Brazilian context, the change in VL transmission pattern [36], observed since 1980's. The geographical distribution of VL has expanded with increasing urbanization. Originally an endemic disease of rural areas and focal occurrence, mainly in the Northeast region of Brazil, VL is

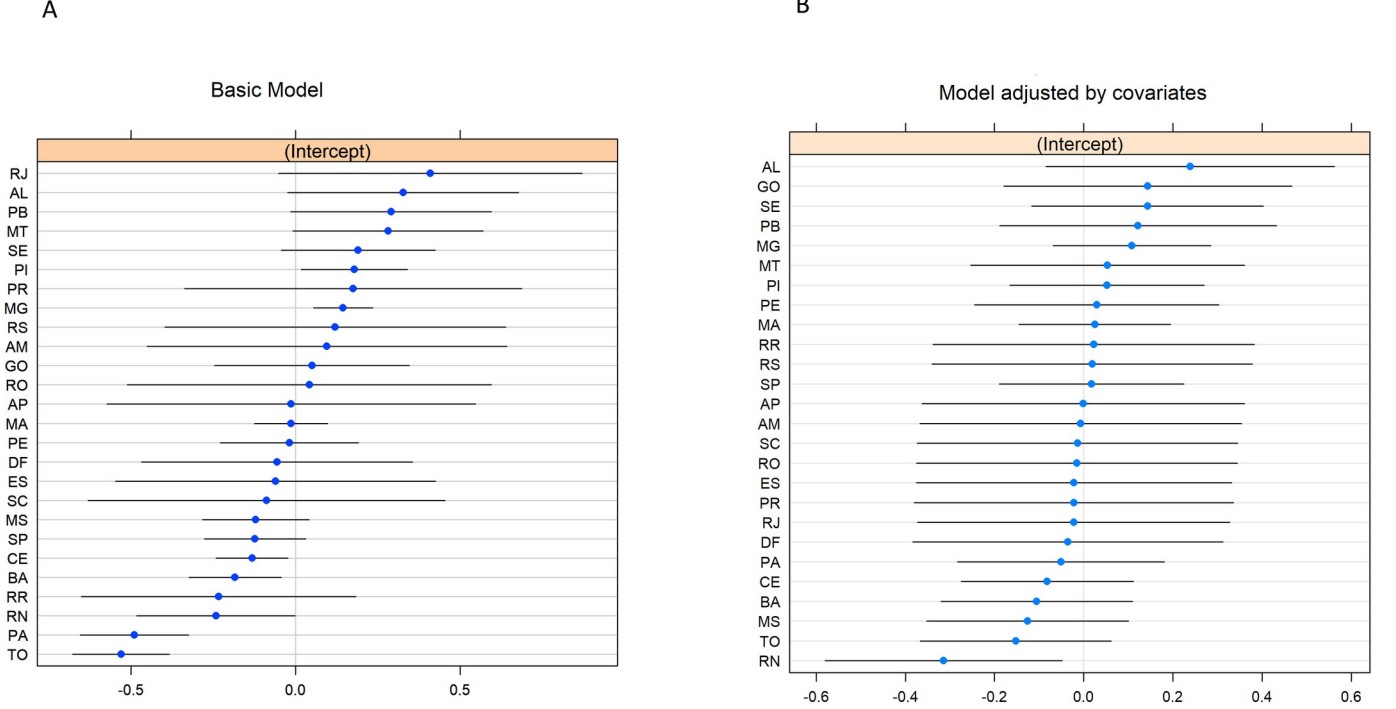

**Fig 6.** Residual effect of VL case-fatality for each Brazilian FU, according to the basic model (left) and model adjusted by covariates (right).

continuously expanding to non-endemic areas, towards the urban centers mainly in the southeast region of the country [37].

Immunochromatographic tests based on K39 antigen (rK39-ICT) represented major progress in VL worldwide in the last years. Endowed with good performance and ease of execution, in theory, rK39-ICT should guarantee the desired speed for VL diagnosis. The ease of execution without sophisticated laboratory resources combined with high performance should bring access to diagnosis to the most remote areas, allowing the immediate initiation of specific therapy. Late diagnosis has been identified in the literature as a factor directly related to death from VL [38–40]. Since VL is a fatal disease if left untreated, the progressive worsening of the clinical condition towards death seems to be an undeniable paradigm. The question that remains unresolved, however, is which, and to what extent, an intervention on a factor associated with the risk of dying has impact on the outcome of a multifactorial disease such as VL. Several factors are related to the delay in diagnosis, but essentially involve the ability of the health care system to recognize the disease and the availability of diagnostic resources. In this sense, both, trained health professionals and the availability of accurate tests are essential conditions for a timely diagnosis. After diagnosis, the stratification of clinical severity and both, specific and supportive treatment, are essential steps. The decentralized diagnosis strategy could, paradoxically, bring VL into the context of low complexity health unit, which may be insufficient for the VL approach except if an efficient flow of referral cases according to severity exists.

In 2009, the first rK39-ICT was incorporated into the Brazilian Leishmaniasis Control Program as an alternative to immunofluorescence antibody test (IFAT) for the diagnosis of VL. However, the IFAT, even with its performance and operational limitations, was still the most widely used serological diagnostic test in Brazil until 2013, when 46% (1586/3470) of VL cases were confirmed using this test. In the same year, 1881 (54%) patients did not undergo any serological examination, and 1509 (43%) VL patients underwent bone marrow parasitological examination during investigation. After 2014, although IFAT was used in the same 46% of confirmed VL cases, according to the Brazilian diseases reporting system (SINAN), another serological test, assumed as a rapid test, started to be performed in more than 50% of VL cases. No change in the proportion of cases with VL laboratory confirmation, around 87%, stable between 2007 and 2017. However, a progressive reduction in the use of parasitological examination was observed in the period, starting from 51% in 2007 to 34% of VL cases in 2017. These data confirm a progressive incorporation of rapid tests for VL in Brazil, apparently replacing the parasitological exam. However, around 50% of confirmed VL cases tested still represents an indicator of underutilization of this tool, designed as the initial screening for every suspected VL case. It is important to note the association between death and VL diagnosis confirmed by parasitological test. Although it may indicate a subgroup of more severe patients, with other alternative diagnoses requiring invasive investigation, this observation may also indicate the use of a time-consuming diagnostic approach, in comparison to serological tests, which may have reflected in the delay in confirming the VL diagnosis and beginning of treatment. This finding reinforces the failure of the VL rapid tests distribution strategy in Brazil, probably due to implementation planning problems and not to the ineffectiveness of the intervention. Several factors could explain this observation, such as the long periods of unavailability of tests, purchase of commercial tests requiring serum and non-digital capillary blood for some periods and finally, a centralized tests distribution strategy, based in the reference laboratories in the FU, and not in primary health care units.

Another disappointing finding is the lack of impact on VL case-fatality of the progressive availability of liposomal amphotericin B in Brazil. Treatment toxicity is classically hypothesized as one of the causes of death in VL. Although all drugs available for leishmaniasis

treatment imply some kind of toxicity, antimony derivatives are considered the drug with the greatest potential to cause serious adverse events, including a fatal one, related to pancreatitis and occasionally severe cardiotoxicity, manifested by prolongation of QTc interval, ventricular tachycardia, Torsade's Pointes, ventricular fibrillation and cardiac arrest. In India, antimony induced cardiotoxicities have been reported in about 10% patients, and mortality attributed to drug related cardiotoxicity is estimated in 5.9% [41,42]. An important historical landmark in relation to changes in the recommendations for the treatment of VL in Brazil was 2010, when the World Health Organization (WHO) established an agreement with the manufacturer Gilead Sciences to ensure a significant reduction in the price of liposomal amphotericin B in countries where VL is endemic. After this agreement, liposomal amphotericin B was incorporated into the Public Health System in Brazil. From 2007 to 2010, less than 10% of the VL cases were treated with liposomal amphotericin B, a percentage that reached 12% in 2011, with a progressive increase until the end of the period, reaching 30% in 2017. In parallel, the percentage of cases treated with meglumine antimoniate was from 74.5% in 2007 to 47% in 2017.

Despite a significant increase from 10 to 30%, it is a percentage that demonstrates a still sub-optimal use of liposomal amphoteric B in Brazil, since solely the VL cases among individuals over 50 years old would represent 30% of treatments. The explanation for this use in a small proportion of the cases is found in the drug distribution strategy adopted by the Ministry of Health, based on a strict clinical criterion slowly expanded. According to the 2006 guide [43], access to liposomal amphotericin B was guaranteed only for patients who have experienced therapeutic failure or toxicity to amphotericin B deoxycholate, kidney transplant recipients or patients with heart or renal failure. From 2011 [44], patients over 50 years, kidney, pregnant women; heart and liver transplant recipients were included in the amphotericin B use criteria. And finally, from September 2013, the criteria were expanded including children up to 1 year and adults over 50 years; patients with higher clinical-laboratorial severity scores; renal, liver or cardiac failure or transplant patients or those presenting prolonged QTc interval or users of drugs that alter the QT interval; presence of hypersensitivity to antimonial derivatives; HIV infection; comorbidities that compromise immunity and previous therapeutic failure. Finally, in addition to the limited amphotericin B use criteria, these numbers also suggest a lack of adherence of prescribers to the Brazilian Minister of Health recommendations.

Regarding the contextual factors, the FU of origin of VL cases with fatal outcome exhibited lower VL incidence and spends more time until the VL notification. One possible explanation for this delay could be the inability of the health system in recognizing a local unusual disease, which seems especially harmful for HIV co-infected patients and for those with low education level, groups more vulnerable to progressive VL complications, either due to issues related to the immune response or the lack of access to health services. The link between access to diagnosis and health facilities and VL case-fatality is also suggested by the inverse association found in univariate analysis between availability of emergency beds, FHP teams and medical doctors and death from VL, which suggests that the robustness of the health system is directly related to the VL clinical outcome.

The higher case-fatality among HIV co-infected VL patients is in turn well demonstrated in Brazil and in other countries where the two infections are endemic. Recently, Elkhoury and colleagues [45] have demonstrated that time of symptoms up to VL reporting for HIV infected individuals was longer than for non-HIV infected patients, which could be explained by the existence of numerous opportunistic conditions with clinical manifestations similar to VL, and HIV itself. The other case-fatality marker found was low schooling, a condition previously linked to VL mortality. There are several explanations for these findings: low schooling as a marker of low socioeconomic status and therefore limiting access to the health system, possibly greater risk of other comorbidities and malnutrition or even the association with HIV

infection. The relationship between educational level and HIV is an interpretation challenge, as long as there are very different realities across countries, and it is changing over time [46]. Recently, schooling is more likely to be associated with a lower risk of HIV infection than earlier in the epidemic. In contrast, in Northeastern Brazil [47], the incidence of VL decreased, while VL-AIDS increased. In addition, HIV infection was confirmed as associated with higher levels of schooling and evidence of higher socioeconomic status.

In the perspective of this nationwide and long-time study, HIV-coinfection and low schooling were related to death, suggesting that the two trends have not canceled each other, on the contrary, they may have been synergistic.

In the present study, interaction between sex and age variables was confirmed. Apparently, the risk of death is higher among young females and among adult men, an observation previously described [14]. At this point, no known explanation for these differential higher risks according to gender is available, which could be related to genetic or social underlying factors.

As an indicator of social and economic development, access to garbage collection, raised as a possible protective factor against the risk of dying from VL. In contrast, no association could be confirmed with other basic infrastructure services, such as access to sewage and water supply. This difference may be related to the possibility of clandestine access to these services, common in irregular occupations in the periphery of large urban centers, or may represent a true positive relationship, as suggested by the urbanization rate association with VL case-fatality, given that water supply and sanitation are services available in urban centers, which are the current VL focus on Brazil. In turn, higher rates of elderly people and life expectancy, both related to a greater number of elderly people in population, were directly related to the risk of dying from VL.

It should be noted that this analysis was not based on a primary clinical database but using the national reporting system based on the transcription of clinical information, often carried out by administrative professionals without health training, with missing and inaccurate information, especially those referring to clinical manifestations and treatment.

For many contextual variables, the analysis was carried out for values ranges with linear relationship with outcome, which does not rule out the influence of these same factors, acting similarly or differently, in contexts not represented by the selected ranges. It is especially relevant for variables that have suffered truncation, such as number of hospital beds, time from symptoms to notification and number of FHP teams. Despite the loss of 21% of the confirmed VL cases in Brazil, in the period, due to lack of available information on the clinical outcome, considering that the presence of VL diagnosis registered in the death declaration triggers mandatory investigation by the epidemiological surveillance system in Brazil, we believe that the number of fatal VL cases losses was negligible and that these results are not likely to be subject to selection bias. In addition, 32% of the cases with unknown clinical outcome were cases transferred to another municipality, which must be reported in duplicate, that is, they are included in the database. Even that, these observations are relevant because they represent a large set of VL data over more than a decade of surveillance by a comprehensive public disease control program. In summary, the results confirm that despite the efforts made in VL approach in Brazil over recent years, the main goal of reducing VL case-fatality has not yet been achieved. Possibly problems in the implementation strategies of two acquisitions of the Brazilian Leishmaniasis Control Program, the rapid tests and liposomal amphotericin B, justify the lack of impact on the VL case-fatality, an experience different from other countries. The variability in the installed structure of the public health system, expressed by the number of emergency beds, medical doctors and health multi-professional teams, all adjusted by population, in addition to the ability of the health system to recognize the disease, which is directly related to the regional incidence of VL, appear as additional markers of lower risk of death. On

the other hand, the number of health units and hospital beds, both absolute indicators, were not related to reduction of the VL death risk, a disagreement possibly due to the fact of they are not related to the quality of the health system, but rather to the absolute size of the population covered.

Despite of none of the contextual factors (FU-level variables) were confirmed as major determinants of the variation in VL case-fatality in Brazil, some of them seems be at least indirectly involved to the fatal outcome, in addition to other still not recognised, acting on the VL risk of death within the spatial unit FU. The non-confirmation of a spatial structure based on the Brazilian FU organization, more than that, the presence of an unstructured spatial influence, show that not only the VL case-fatality rates are not organized in FU conglomerates with similar rates, but there is significant variability dependent mainly on intra-FU factors. These results suggest that the geopolitical structure represented by the FU are insufficient to reflect the differences within the unit, since the magnitude of the effects of contextual variables were generally weak, losing their importance in multiple models. Influence of contextual factors should be explored, in future studies, using smaller spatial scales of analysis, possibly so, differences within socio-economically developed regions, but permeated by pockets of poverty and lack of assistance, can be captured by the model, and the influence of differences in access to health, professional training, as well as the influence of socio-sanitary and epidemiological aspects could emerge.

For now, these observations point out new windows of opportunities in terms of public policies to be implemented to achieve reduction in VL case-fatality. In particular, they suggest that improvement in VL indicators depends on the strengthening of the Brazilian Unique Health System as a whole and part of the resources need to be directed towards capacity building, including services and professionals.

## Supporting information

**S1 Table. Confirmed visceral leishmaniasis cases according to FU, Brazil, 2007 and 2017.**
(DOCX)

**S2 Table. VL incidence by FU, between 2007 and 2017.**
(DOCX)

## Acknowledgments

GC is grateful for Universidade Federal de Minas Gerais (UFMG), for the training license, and for Department of Epidemiology, Center for Public Health, Medical University of Vienna, Austria for the invitation as visiting researcher fellow, between December 2019 to January 2020, when part of this study was developed.

## Author Contributions

**Conceptualization:** Gláucia Cota, Taynãna Cesar Simões.

**Data curation:** Gláucia Cota.

**Formal analysis:** Gláucia Cota, Taynãna Cesar Simões.

**Methodology:** Gláucia Cota, Astrid Christine Erber, Eva Schernhammer, Taynãna Cesar Simões.

**Writing – original draft:** Gláucia Cota.

**Writing – review & editing:** Gláucia Cota, Astrid Christine Erber, Eva Schernhammer, Taynãna Cesar Simões.

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
