## [Decision Letter · Decision Letter 0]

27 Nov 2020

Dear Dr Cota,

Thank you very much for submitting your manuscript "Inequalities of visceral leishmaniasis case-fatality in Brazil: a multilevel modeling considering space, time, individual and contextual factors" for consideration at PLOS Neglected Tropical Diseases. As with all papers reviewed by the journal, your manuscript was reviewed by members of the editorial board and by several independent reviewers. In light of the reviews (below this email), we would like to invite the resubmission of a significantly-revised version that takes into account the reviewers' comments. 

We cannot make any decision about publication until we have seen the revised manuscript and your response to the reviewers' comments. Your revised manuscript is also likely to be sent to reviewers for further evaluation.

Sincerely,

Alberto Novaes Ramos Jr, M.D., M.P.H., Ph.D.

Guest Editor

Nadira Karunaweera

Deputy Editor

Reviewer's Responses to Questions

**Key Review Criteria Required for Acceptance?**

**Methods**

-Are the objectives of the study clearly articulated with a clear testable hypothesis stated?

-Is the study design appropriate to address the stated objectives?

-Is the population clearly described and appropriate for the hypothesis being tested?

-Is the sample size sufficient to ensure adequate power to address the hypothesis being tested?

-Were correct statistical analysis used to support conclusions?

-Are there concerns about ethical or regulatory requirements being met?

Reviewer #1: Adequate methods have been used in the analyses. 

However, the original data (national) seems to be of low quality, particularly those regarding the clinical data. Often, non-qualified personnel extracts the data from the clinical files in health units. It is noteworthy the lousy quality of the clinical descriptions by physicians in the country. The original data's low quality may have hampered the data analyses and reduced the conclusions' credibility. The authors should address this point in the discussion. 

It might also be better to standardize the proportion of deaths among the patients with the disease since the authors used both case-fatality (more appropriate) and lethality.

Reviewer #2: Yes

Reviewer #3: I found the objectives to be clearly stated in the Abstract (lines 29-31), but suggest minor revision to more clearly restate/elaborate at the end of the Intro (127-131).

The study design with respect to the temporal trend and spatial dependency in case fatality is generally strong, and I have only minor methodological suggestions. But with respect to assessing determinants of lethality at the individual and regional level, I found the methodology (and results) to be incompletely described or ambiguous in quite a few places. When I read the Methods section on its own I had only small suggestions/questions. But when I read the Results I had substantial difficulty mapping stated results back to described Methods. I'll describe most of these issues later in the Results section.

** Major suggestions relating to Methodology: **

(1). The distinction between "federated unit (FU)" and "FU-by-year" groupings should be clarified. Throughout the text there are references to two categories: FU with below-median case fatality and FU with above-median case fatality. But at a few points in the text it seems like the actual split for analysis purposes was 'FU-by-year combinations below the national median' vs. 'FU-by-year combinations above the national median'.

E.g. Line 360: "Some FUs can appear in both groups in Fig 3 since the FU's position in relation to national lethality, in some cases, was not stable over the period 2007-2017." (In fact I count 18/26 FU in Fig. 3 that appear in both the above-median and below-median groups).

I don't think it's necessary to split FU into groups at all (see #2), but if the split is retained I think it would be preferable to actually split by FU (based on FU-specific fatality across all study years) rather than FU-by-year. The latter (i.e. current) approach might be prone to statistical artifacts due to varying year coverage for different FU within each model.

(2). I don't see any reason to analyze the below-median FU and above-median FU using separate models (or perhaps below-median and above-median *FU-years*, see above) (e.g. lines 218-225). It's of course possible that some factors have non-linear effects on case fatality, but I can't think of a priori reasons for such effects in most of the variables analyzed here, and there's little relevant discussion in the manuscript. Indeed the reported odds ratios are almost always similar between the above-median and below-median groups (Table 3).

Splitting on the median seems arbitrary and could have undesirable side effects:

- Halving the sample size in each model reduces the statistical power to detect 'real effects'

- Using two models instead of one increases the likelihood of false-positives, particularly given the 'forward selection on many variables' approach used here

Perhaps the biggest challenge is that it's not clear what to make of results that apparently differ between the two categories (e.g. HIV-coinfection, education, and hospital beds/capita are suggested to only affect cases fatality in above-median FU). Is the lack of a significant result in the below-median group a false negative due to lack of statistical power? Or the above-median result a false positive? Or does the effect of these variables on fatality really change as a function of FU-specific fatality levels?

I think a single-model approach would be preferable here, and much simpler to present and interpret. If the effect of any given variable on case fatality appears non-linear, it could be modelled using e.g. a polynomial term, a GAM smoother, or investigated using supplementary analysis on FU subsets.

** Minor suggestions relating to Methodology: **

188: "adjusted" is not quite right here. Suggest using something like "generalized additive models (GAMs) were fit..." instead.

188: Is it necessary to use a GAM rather than the simpler GLM here? The time-series in Fig 1 looks reasonably linear to me, and the straight GAM fit line (Fig 1 right) seems to indicate zero smoothing. Also the single temporal coefficient presented in the results (4% per year, line 277) seems to represent a GLM coefficient rather than a GAM coefficient, no? If sticking with GAM, clarify the smoother type (thin plate, penalized cubic, etc.).

196-199: If the ultimate CAR model uses counts of deaths as the response and counts of cases as an offset, the Empirical Bayes smoothing seems unnecessary to me. I.e. The higher uncertainty in FU with low case counts is already explicitly incorporated into the CAR model with the offset term. If retained, the Empirical Bayes methods should be described in more detail (software used, model structure, settings, etc.).

201-204: Suggest clarifying the structure of the non-spatial term. Random intercepts?

205-207: Suggest rephrasing to clarify that the offset is the number of confirmed VL cases. Also, don't need to put "offset" in quotes here as the term has already been used at line 189.

214: Clarify the difference between the "random term at the FU level" and the "random intercept at the FU level" described in the previous sentence. Does the "random term" mean a 'random slope' term for each FU with respect to year or some other variable?

227: Clarify which "parametric or non-parametric hypothesis tests" were used, the software, etc. Also, are these results presented anywhere (perhaps these are the 'Univariate' results in Table 3?). Otherwise it's not clear what the purposes of these tests was.

229: Clarify how multicollinearity was tested for, which variables were affected, and what actions were taken when it was observed.

248: This line indicates that analyses were done in R and SPSS, but apart from the CAR model (Line 209) it's not clear which software/packages were used for which statistical analyses.

**Results**

-Does the analysis presented match the analysis plan?

-Are the results clearly and completely presented?

-Are the figures (Tables, Images) of sufficient quality for clarity?

Reviewer #1: Figures are hard to be read due to the low resolution, and there is no apparent correspondence of the figures with their legend. This fact has hampered the review quality.

Reviewer #2: Yes

Reviewer #3: As noted above, particularly with respect to the analyses on determinants of lethality at the individual and regional level, I had substantial difficulty linking the stated results back to the described Methods.

** Suggestions relating to Results: **

1. It's not clear to me what "Univariate analysis" and "Multiple analysis" refer to (e.g. Table 3). Normally I would think "Univariate" refers to independent tests between fatality and each separate variable (whereas "Multiple" refers to the multivariate models obtained through forward variable selection, i.e. 232-248), but I don't see the univariate tests described anywhere in the Methods. Also, lines 244-246 make me think random coefficients are a final step in the forward selection (i.e. "Multiple analysis") process, but then there is a random-coefficient term in the Univariate column of Table 3 (though based on Lines 343-345 perhaps this term was placed in the wrong column?).

There's also not enough description of the apparent difference between Univariate/Multiple results. What does it mean that the best Multiple model for above-average FU contained only a "Report year" term and no other variables? And that the odds ratios for "Report year" are below 1 in the Multiple analysis but above 1 in the Univariate?

2. It's not always clear how the stated results relate to the presented statistics. E.g. Why are HIV-coinfection and education level stated to be linked to case fatality only in above-median FU (337-339; also noted in the Abstract and Discussion)? In Table 3 the relevant odds ratios look similar for both the above-median and below-median groups. Similarly, the Discussion (520-522) notes the absence of an effect of 'hospital-beds/capita' on case fatality, but again it's not clear to me where this result comes from given the similar odds ratios in Table 3.

3. Lines 343-345, 354-356, 356-359: I think the results relating to random coefficient terms are generally misinterpreted (also in the Abstract). The 'hospital beds/capita' term seems to be a 'random coefficient/slope' with respect to FU (i.e. Fig 3 left). In that case, the inclusion of this term through variable selection indicates that the slope of the relationship between 'hospital beds/capita' and case fatality varies by FU. E.g. Based on Fig 3, in federated unit "AL" the slope of the relationship between 'hospital beds/capita' and case fatality is relatively more negative than the average (i.e. the fixed effect) and in "MA" the slope of the relationship between 'hospital beds/capita' and case fatality is relatively more positive than the average. 

4. Lines 363-365: Suggest rephrasing. The lack of a significant FU effect in the low-lethality group doesn't necessarily mean "case fatality can be explained by factors at the individual level". Unexplained variation in the model could be occurring at other levels (i.e. any level between individual and national except FU).

5. For each model selection process, suggest clarifying in the Results which variables made it into the final model. Is it all the variables in Table 3 with non-missing odds ratios? Or are the odds ratios given only for variables with p < 0.05? Relatedly, it's not clear to me why there aren't odds ratios given for every variable in the Univariate columns (perhaps related to #1). Also need to better explain the process for dealing with multicolinearity (229-230, 241-243), and which (if any) variables were de-selected as a result. Surely some of the variables in Table 3 are highly correlated, no?

6. Clarify levels/units of each variable (e.g. Table 3). For instance, giving an odds-ratio for "Gender" doesn't indicate which direction the effect is in. And what are the possible values of e.g. "Education level"? The single odds ratio makes it seem like it's a continuous variable here?

7. Make sure to report the complete test results for statistical tests (usually test statistic, degrees of freedom, and p-value). E.g. Table 2 (Results) gives bare p-values and I can't find any details on test type within the text.

8. Lines 339-341, 354-356: Clarify the metric being used to compare the relevant influence of different variables.

9. Make sure figure axes have meaningful labels. E.g.

- Fig. 1 right, y-axis: "Case fatality (logit scale)" (or transform to probability scale)

- Fig. 2 middle, x-axes: "Federated unit (integer index)"

- Fig. 3, x-axes: "Estimated model coefficient (case fatality on logit scale)"

** Other minor points related to Results: **

278: Clarify the time unit. 4% increase per year or over the 11-year period? Would be good to add confidence interval for the trend estimate here too.

286-290, 301-304: The caption and in text-references to the middle panel of Fig 2 are reversed compared to the figure, which shows uf.nu on the left and uf.theta on the right.

292-293: Clarify the evidence for "good fit regarding the estimates of hyperparameter precision".

296: The use of "adjusted" is a bit confusing here. Would be clearer to say something like "Based on this result, we included FU-level random intercepts in our multi-level models of lethality described in the following section".

297-299: This line on temporal effects seems out of place in the spatial section. Suggest moving to temporal section instead.

**Conclusions**

-Are the conclusions supported by the data presented?

-Are the limitations of analysis clearly described?

-Do the authors discuss how these data can be helpful to advance our understanding of the topic under study?

-Is public health relevance addressed?

Reviewer #1: The authors should be clear that the policy changes to VL's diagnosis and treatment in Brazil have worsened case-fatality and not use the euphemistic wording that they had not improved it.

Possible explanations are that patients diagnosed by rapid tests are usually at the primary care units. There, the use of antimonials is much more likely due to the national recommendations themselves, and severe cases may not be identified and forwarded to secondary- or tertiary units and not adequately treated.

Another point is that any clinical trials have not evaluated the treatment of severe cases of VL up to date.

Reviewer #2: Yes

Reviewer #3: (No Response)

**Editorial and Data Presentation Modifications?**

Reviewer #1: For the paper quality, improvement in data interpretation and conclusions are necessary.

Reviewer #2: - Abstract: several findings are only mentioned first in the conclusion, would recommend to put these in the “Findings”

- summary: In the abstract hospital beds is mentioned as factor but not in the summary

- line 62: “trends and factors – recommend to add “and”

- line 126: justified: suggest to replace by “explained” – also in the discussion

- line 347: in contrast to another: suggest to replace “another” by “the other”

- line +516: lack of diagnostic confirmation: does it mean not test was done or that the test could be negative but the patient was still diagnosed on clinical grounds?

Reviewer #3: - 79: what's the time period over which the 200k-400k cases were added?

- 97: consider "Americas region" instead for clarity

- 121/126: consider "explained" instead of "justified"

- 149: "accessed" instead of "assessed"

- 162: more common to describe Methods in past tense ("will also be considered" -> "were also considered")

- 340: consider "influential" instead of "influent"

- Table 1 (Methods): consider reference/footnote for race categories (e.g. IBGE), as terminology might vary by country

- Table 2 (Results): "percent" instead of "percentual"

- Table 3 (Results): use "." as decimal separator for consistency with rest of manuscript

**Summary and General Comments**

Reviewer #1: This is a relevant and well-done study that should be published. However, changes in the discussion and conclusions seem to be appropriate.

Reviewer #2: This is a useful paper on an important topic. The data are clearly presented, the analysis was conducted carefully and the findings are nicely discussed.

I only have a few minor points – all discretionary

Reviewer #3: I think the study objectives are important, and the data sources used are well-suited to addressing them. The analyses with respect to temporal trend and spatial dependency in case fatality from visceral leishmaniasis are generally sound and well-presented. However, there are major shortcomings in the description of the methodology and results for the analyses on individual- and regional-level factors impacting case fatality, which make it difficult to assess the quality of those analyses and interpretation of results.

PLOS authors have the option to publish the peer review history of their article (what does this mean?). If published, this will include your full peer review and any attached files.

Reviewer #1: Yes: Carlos H N Costa

Reviewer #2: No

Reviewer #3: No
---

## [Decision Letter · Decision Letter 1]

13 Feb 2021

Dear Dr Cota,

Thank you very much for submitting your manuscript "Inequalities of visceral leishmaniasis case-fatality in Brazil: a multilevel modeling considering space, time, individual and contextual factors" for consideration at PLOS Neglected Tropical Diseases. As with all papers reviewed by the journal, your manuscript was reviewed by members of the editorial board and by several independent reviewers. In light of the reviews (below this email), we would like to invite the resubmission of a significantly-revised version that takes into account the reviewers' comments. 

We cannot make any decision about publication until we have seen the revised manuscript and your response to the reviewers' comments. Your revised manuscript is also likely to be sent to reviewers for further evaluation.

Sincerely,

Alberto Novaes Ramos Jr

Associate Editor

Nadira Karunaweera

Deputy Editor

Reviewer's Responses to Questions

**Key Review Criteria Required for Acceptance?**

**Methods**

-Are the objectives of the study clearly articulated with a clear testable hypothesis stated?

-Is the study design appropriate to address the stated objectives?

-Is the population clearly described and appropriate for the hypothesis being tested?

-Is the sample size sufficient to ensure adequate power to address the hypothesis being tested?

-Were correct statistical analysis used to support conclusions?

-Are there concerns about ethical or regulatory requirements being met?

Reviewer #1: Methods are excellent, well designed and well applied.

Reviewer #3: 1. Most of my comments related to methodology from the previous round of revision have been addressed with this revised version. However, I have some new concerns regarding the new analyses.

Much like the approach of splitting FU into below-median and above-median groups in the original analysis, the approach of discretizing the continuous variables in this revised version seems arbitrary and subjective, and the methodology is not defined clearly enough to be reproducible. E.g. For variable 'Age (years)', why is there no split at the change-point around age 10 (Fig. 5b)?

Regardless of how the split-points are chosen, the process of discretizing continuous variables seems unnecessary, and reduces the information available. E.g. The discretized analysis with age only tells us that the >=20 group experiences higher CFR on average than the <20 group, whereas if we retain age as a continuous variable we see that CFR closely mirrors the typical 'bathtub-shape' human mortality curve — decreasing from birth to age ~10, then progressively increasing with age thereafter.

Likewise, the discretized analysis of onset to reporting delay suggests that CFR is ~6x higher in the >=200-day group compared to the <200-day group, but it's not clear how meaningful this is considering that the 200-day split point is extremely high in the distribution of reporting delay (97th percentile based on data in Table 2). It's hard to tell without seeing the raw data, but the spline relationship between CFR and reporting delay looks relatively linear, in which case it would be more meaningful to interpret a linear coefficient (e.g. X% increase in CFR for every additional week of delay).

Instead of the discretized approach, you could simply use linear terms for the continuous variables, perhaps using transformations (log, square-root, etc.) for variables where the linearity assumptions are severely violated. 

Alternatively, you could use splines for all analyses. E.g. For the univariate analyses you could compare a spline on the variable of interest (x) to the appropriate null model, using likelihood ratio tests or AIC, e.g.

m1 <- gam(outcome ~ s(x) + year + s(FU, bs = "re"), family = "binomial")

m0 <- gam(outcome ~ year + s(FU, bs = "re"), family = "binomial")

anova(m1, m0, test = "LRT")

Admittedly this approach could be difficult for the multivariate analysis, as complex spline models with many variables might not converge.

Regardless of the approach, I would strongly suggest adding the actual data points to plots of CFR vs explanatory variables (e.g. Fig 5), so that readers can assess the underlying distributions/relationships.

2. There are methodological aspects of the multilevel modelling that are still unclear, and I'm not confident that another author with the same data could reproduce this analysis without further information. E.g.

- 250: The use of "subsequently" is confusing here. Is this describing the multilevel modelling, and suggesting that FU-level variables were tested and added prior to individual-level variables? Presumably this is not the case, given that no FU-level variables made it into the final model (Table 3). 

- 259-260: This line indicates an alpha = 20% acceptance threshold for "variables at the individual level", but what about the FU-level variables? Do they have a different threshold?

- 262: I don't understand the meaning of "with fixed coefficients (varying between FUs)" here. I think the FU-level random intercepts and random coefficients are the only parameters that vary between FUs, no?

- 266: What was the acceptance threshold for comparisons based on AIC? Also, based on the described methods, I'm not clear on which comparisons would involve non-nested models. Is this only for the random coefficients?

Other points related to methodology:

119: Isn't the offset the number of confirmed cases *with known clinical outcome*, rather than the total number of confirmed cases per se? On this point, given that only 79% of confirmed cases have known clinical outcome (line 287), I think a very brief discussion of possible biases in missing data would be appropriate (probably in the Discussion section). E.g. Is it possible the ~20% of cases with unknown outcomes are biased toward survivors, or exhibit temporal trends in fatality different from the 79% with known outcomes?

195: It still seems unnecessary to use a GAM here. The temporal-trend results later presented are based on a GLM (Table 2, line 303), and the temporal term in the univariate and multivariate analysis is likewise a linear term. Why not just use the linear term (GLM) for all temporal analyses?

215, 234: I think "expected VL cases" should instead be "observed VL cases"? "Expected" implies a prediction or a model-estimate, but I think the offset is just the observed number of cases per FU.

243: "greatest association" based on what metric?

259: Related to point #1 above, it's not clear to me why the discretization should aim to split a variable into segments where the relationship with the response is linear, per se. Within-factor linearity is not a model assumption. As noted above, I suggest omitting the discretization step, but if retained, this particular approach should be better justified/explained.

**Results**

-Does the analysis presented match the analysis plan?

-Are the results clearly and completely presented?

-Are the figures (Tables, Images) of sufficient quality for clarity?

Reviewer #1: Very relevant results, likely influencing policy making.

Reviewer #3: 285-286: Clarify the level that the average/range are taken across (e.g. years, FU, or both).

304: The 95%CI uses a comma decimal separator whereas the rest of the text primarily uses a period (though there are also comma separators for incidence and case fatality in Table 1).

318: Clarify "has a low probability of occurrence in its a posteriori distribution".

335: The subheading "Individual level analysis" in the Results section corresponds to subheading "Multilevel analysis" in the Methods section. Suggest using a common subheading. "Multilevel analysis" makes more sense.

337: Not clear what the significance of "an age up to 14 years" is here. Could just omit and report only the median/IQR (16 years, IQR 2-39).

338: A 75% IQ is unconventional. Is this instead the 50% inter-quartile range, i.e. 25-75%?

341: Consider changing "totalized" to "totalled"

347: The reported median age of the cured group (13 years) does not match the 95% CI (25-75 years).

345-348: The reported medians and 95% CIs in onset to treatment delay in the cured (33, 16-76 days) and fatal groups (30, 15-59 days) are almost certainly *not* consistent with a significant difference between the groups. Perhaps the 95% CI are actually IQR?

350: Add full test results anytime a p-value or claim of significance is made. Generally this includes test static, degrees of freedom, and p-value. Could also note the test type if it's not clear from the Methods. Also, should use e.g. p < 0.001 rather than p = 0.00.

361: Clarify the correlation metric used here (presumably R^2).

384: This odds ratio (6.4) seems surprising given the data in Table 2 (I calculate a raw odds ratio of 1.13; CFRs: 8% vs. 13%). Of course, the ORs in Table 2 are based on a model with year and FU-intercepts, so could be different. Just worth double checking.

388-390: I'm not clear on why these 3 variables (race, locale, case classification) aren't presented in Table 2. Perhaps because they are statistically non-significant? Would be preferable to present results consistently regardless of significance level.

399: Should say "Fig 5l" instead of "Table 5l"

411-412: The text and Table 2 imply that CFR declines with more medical doctors per capita, but Fig 5i suggests it generally increases.

437: The use of "multiple hierarchical analysis" in the caption for Table 2 is confusing given that the models are described as univariate. Could change to something like "generalized linear mixed models with FU-level random intercepts".

Table 2: I don't understand the meaning of "VL cases (%) according to..." in some of the variable labels. I think can just be omitted?

Fig 1. The use of the two scales (deaths vs. CFR) but single axis in the left panel is awkward. And the right panel should be transformed to the scale of the response (i.e. probability or percentage) and given an informative y-axis label instead of "spline(year)". Also, isn't the right panel showing a temporal effect on VL case-fatality instead of on "VL Deaths" per se (given that the model uses cases as an offset term)? I think it would be preferable to show the death numbers in one panel and the CFR raw data (red line) and model predictions (spline panel but on the response scale) together in a separate panel.

Fig 2. The breaks for the colour scale in the right panel seem to be truncated too low (15%) — only about halfway up the scale.

Fig 3. Add axis scales and labels for panels (a) and (b). Also, just an aesthetic point, try to make panel (d) the same size as (c).

Fig 5. Consider standardizing the range of the y-axis so that it's possible to visually compare the relative magnitude of the effect size across panels/variables. Also, the y-axes should be transformed to a meaningful scale (i.e. CFR, probability or percentage). The current label "spline" is not informative. Also, not clear why the x-axis data is depicted as dashes along the bottom in some panels but not others. Regardless, I'd strongly suggest adding the actual data points (x and y) instead to every panel so readers can access the fit of the splines.

Fig 5c. The panel depicts a split-point at 1400 days (also noted line 391), but Table 2 suggests this variable was treated as a continuous linear predictor.

Fig 6. What is nm_uf? Wasn't this a parameter of the spatial model? Also, suggest consistent use of subheadings — e.g. why use a subheading (Basic Model) for panel (a) but no subheading in (b)? Also, add label (including units) to the x-axis.

**Conclusions**

-Are the conclusions supported by the data presented?

-Are the limitations of analysis clearly described?

-Do the authors discuss how these data can be helpful to advance our understanding of the topic under study?

-Is public health relevance addressed?

Reviewer #1: Conclusions are in accordance with the results, albeit debatable.

Reviewer #3: 1. The biggest remaining issue is the presentation/interpretation of results. As far as I can tell, none of the FU-level variables were selected into the final multilevel model. The 7 variables that were selected were all individual-level factors (age, symptoms, gender, etc.) (Table 3).

To me the simplest interpretation of this result is that, after accounting for individual-level factors, *none* of the FU-level variables (# hospital beds, garbage collection, proportion children, etc.) account for a substantial proportion of variation in case fatality.

The authors do note this caveat in the Discussion (633-636)

"These results suggest that the geopolitical structure represented by the FU are insufficient to reflect the differences within the unit, since the magnitude of the effects of contextual variables were generally weak, losing their importance in multiple models."

but nonetheless suggest that some of the FU-level variables are important predictors of case fatality in other parts of the manuscript, e.g.

- "...unavailability of emergency beds and health professionals (the last two only in univariate analysis) were identified as related to VL death risk." (line 43)

- "Lower VL incidence was also associated to VL case-fatality, suggesting that unfamiliarity with the disease may delay appropriate medical management" (line 45)

- "The indicator of social and economic development, access to garbage collection, was confirmed as protective against the risk of dying from VL." (line 603)

- "higher rates of elderly people and life expectancy, both related to a greater number of elderly people in population, were directly related to the risk of dying from VL" (line 610)

I think these conclusions are overstated given the analyses presented. It could of course be the case that some of the FU-level variables affect CFR indirectly through their impact on individual-level factors. E.g. Perhaps some of the health-related FU variables (e.g. doctors per capita) indirectly affect CFR through their effect on onset-to-treatment delays. But this sort of mediating relationship could/should be tested directly.

Along the above lines, particularly with respect to FU-level variables, I think a more focused analysis would be preferable. Potential causal pathways can be identified from the outset, and the relevant FU-level variables chosen in a more targeted fashion. In the current analysis, I think there's not much to be learned from the inclusion of FU-level "Gender proportion" in the same model as individual-level patient gender, or FU-level "Proportion under 5 years" in the same model as individual-level patient age, etc.

2. Related to the methodological concerns about unnecessary discretization, some of the conclusions regarding FU-level variables (also noted in point #1 above) are not evident when looking at the splines in Fig. 5. E.g. The authors conclude "Lower VL incidence was also associated to VL case-fatality, suggesting that unfamiliarity with the disease may delay appropriate medical management" (Abstract, line 45), but the spline relationship in Fig 5d does not suggest a simple unidirectional relationship between incidence and CFR. The spline relationships between CFR and emergency beds, number of family health programs, and garbage collection are similarly complex.

The most important step to clarify these relationship would be to show the underlying data in the spline plots. Assuming the splines are a faithful representation of those data, the conclusions should be revised to add nuance, and focus on describing the actual relationships, even if they are non-linear and complex, and de-emphasizing the discretized results.

**Editorial and Data Presentation Modifications?**

Reviewer #1: Just change the word case-lethality for case-fatality in line 66. Minor revision.

Reviewer #3: (No Response)

**Summary and General Comments**

Reviewer #1: The article is ready to be published after the minor change.

Reviewer #3: (No Response)

PLOS authors have the option to publish the peer review history of their article (what does this mean?). If published, this will include your full peer review and any attached files.

Reviewer #1: Yes: Carlos H N Costa

Reviewer #3: No
---

## [Decision Letter · Decision Letter 2]

29 Apr 2021

Dear Dr Cota,

Thank you very much for submitting your manuscript "Inequalities of visceral leishmaniasis case-fatality in Brazil: a multilevel modeling considering space, time, individual and contextual factors" for consideration at PLOS Neglected Tropical Diseases. As with all papers reviewed by the journal, your manuscript was reviewed by members of the editorial board and by several independent reviewers. In light of the reviews (below this email), we would like to invite the resubmission of a significantly-revised version that takes into account the reviewers' comments. 

We cannot make any decision about publication until we have seen the revised manuscript and your response to the reviewers' comments. Your revised manuscript is also likely to be sent to reviewers for further evaluation.

Sincerely,

Alberto Novaes Ramos Jr

Associate Editor

Nadira Karunaweera

Deputy Editor

Reviewer's Responses to Questions

**Summary and General Comments**

Reviewer #1: No comments.

Reviewer #2: (No Response)

Reviewer #3: (No Response)

PLOS authors have the option to publish the peer review history of their article (what does this mean?). If published, this will include your full peer review and any attached files.

Reviewer #1: Yes: Carlos H N Costa

Reviewer #2: No

Reviewer #3: No

**Key Review Criteria Required for Acceptance?**

**Methods**

-Are the objectives of the study clearly articulated with a clear testable hypothesis stated?

-Is the study design appropriate to address the stated objectives?

-Is the population clearly described and appropriate for the hypothesis being tested?

-Is the sample size sufficient to ensure adequate power to address the hypothesis being tested?

-Were correct statistical analysis used to support conclusions?

-Are there concerns about ethical or regulatory requirements being met?

Reviewer #3: In my opinion there remains significant problems with the clarity of the methods, and with the methodological approach of discretizing or truncating explanatory variables based on visual assessment of GAM models.

Discretizing/truncation based on GAM:

The most problematic use of discretization/truncation is for the variable Emergency beds, where, as far as I understand, the authors decide based on the GAM model to exclude data above a setpoint of 1000 emergency beds / 100k inhabitants. The resulting negative relationship between emergency beds and fatality (which is repeatedly referenced, e.g. lines 43, 70, 410, 582, 636), is in my view completely meaningless because the authors have specifically chosen only to analyze a segment of the curve in which the trend is negative, while ignoring the next segment in which fatality generally *increases* with the number of emergency beds. This problem of conclusions influenced by arbitrary truncation also applies to the variable "Number of FHP Teams".

More generally, it's still not sufficiently clear how the authors have chosen the split points, or how they decide between discretization (e.g. Age) and truncation (e.g. Emergency beds).

The assessment seems to somehow depend on whether the GAM confidence interval crosses zero, e.g. "... inferring in which ranges of values this association is significant, which can be declared when ranges of values of the curve do not belong to the estimated confidence interval and contain included zero value." (259-261). But it's not clear to me how to get from this description to the specific split points in Fig 5. E.g. For variable Standardized VL Incidence, there are split points at the 2 inflection points on the curve, which seem unrelated to the CI crossing zero. In contrast, for Age there is a split point where the CI crosses zero and *not* at the clear inflection point around age 10. For Family Health Programs, the first split point is at an inflection point and the second occurs where the CI crosses zero.

In the response to the previous review (point #1) the authors suggest "There are countless ways to apply the GAM models for this purpose", and go on to explain several seemingly different approaches, but this did not help me understand which approach was actually used for which variables, or why.

I also think the interpretation that regions of the GAM confidence interval that cross zero indicate "a null effect in that region" (e.g. 259-261, 386-387) isn't quite correct, nor is it meaningful to visually assess regions of a GAM curve where the relationship is "significant". The GAM fit lines are showing the predicted mean value of y (+/- confidence interval) on the transformed scale (e.g. logit) at a given value of x, relative to the overall mean value of y. They're not showing the *effect* of x on y (i.e. the slope of the relationship) at a given value of x (though this can be interpreted from the slope of the fit line). E.g. for variable Medical doctors (Fig. 5i), the GAM confidence interval overlaps with zero in the region of about 0.95-1.2 doctors per 100k inhabitants, but the slope of the relationship between Medical doctors and fatality is still clearly positive (and relatively linear) in this region, so I don't see why/how we would use the fact of an overlapping CI to inform discretization.

Similarly, as I noted in my previous review, "it's not clear to me why the discretization should aim to split a variable into segments where the relationship with the response is linear, per se." I understand that GLM's have a linearity assumption, but if an explanatory variable is discretized prior to analysis there is not an assumption of linearity *within each discrete category* of that variable per se.

I can see the logic of using GAM models to identify linear segments if the purpose is truncation — to exclude values outside the linear segment (though I think think truncation is a bad approach generally). But I don't see the logic to using GAM to identify linear segments if the purpose is discretization. E.g. the variable "Interval between symptom onset and reporting" was dichotomized using a cutpoint of 200 days, which generates two roughly linear segments based on the spline. But an analysis that uses this dichotomized explanatory variable doesn't require that each segment is linear... the analysis only sees two groups, >=200 or <200, with no relation to the continuous variable "Interval between symptom onset and reporting". So why choose split points specifically based on linearity here?

I'm sympathetic to the authors' previous response (point #1) about using discretization to aid in interpretability, but this still requires more nuance and a clearer explanation of the methods than are presented here. Finally, in response to my previous comment about discretization, the authors respond (#39) "... the approach based on the initial assessment of the presence of linearity by GAM models is well established in the literature...", citing the reference section of a book "Generalized Additive Models" and a paper "Generalized additive models for location, scale and shape". I looked through the book and the paper the authors cite, and see no relation to the particular discretization approach used here. I haven't checked every single reference in the book, but the authors can clarify the relevant references if necessary.

Other things to clarify:

244:

- Clarify which p-value is being referred to in the line "those with the lowest p value were selected for the multiple models". Is this the p-value from the univariate models described below (i.e. also adjusting for time, FU, and age distribution), or the "parametric and non-parametric tests" described immediately above (237-243)? The authors' response letter (#13) indicates it's the univariate models but this is not clear in the text itself.

- Should explicitly state within the manuscript or appendix, probably in a table, which variables were excluded on the basis of multicollinearity. And clarify whether these variables were also excluded from univariate analysis. I'm assuming not, since e.g. child mortality and life expectancy are both included in the univariate results Table 2 (Results) despite looking nearly perfectly correlated (Fig 4).

257:

The GAM(M) models used as part of the multilevel analyses should be more fully described here, e.g.

- Error distribution (presumably binomial)

- What type of spline (e.g. thin plate, cubic, etc.). E.g. The default in the spline function s() in the mgcv package is "thin plate regression spline".

- Is a time term included as in the GLMM models?

- Are there random effects included (e.g. FU)? If so how are these specified, as there are multiple possible approaches with GAMMs. The Results section notes both GAM (385) and GAMM (392), so it's unclear where these are mixed models or not.

268:

There are various approaches to stepwise variable selection. This sounds like "forward variable selection", but should clarify explicitly here. I.e. I'm not positive from the description that it's not a bi-direction selection approach, where variables could be added or removed at each step.

269:

Relating to the point about explicitly stating which variables were omitted on the basis of multicollinearity, the line "variables were introduced one by one into the basic model in decreasing order of significance, **considering the multicollinearity and the particularities detected in the previous analyzes**" is not sufficiently clear to be reproducible. It's difficult to judge the exact correlation values from Figure 4, but for example, MedicalDoctors looks potentially highly correlated (|r| > 0.85) with lots of variables: Sewage, LifeExpectancy, Water, Garbage, Urbanization, ChildMortality, and Children<5. Is the exclusion of correlated variables done at the very beginning, such that 7 of these 8 variables would be immediately excluded from all further analyses (all except the one most strongly related to fatality based on the univariate tests)? Or are the exclusions updated at every step such that, e.g., the 7 variables correlated with MedicalDoctors are only excluded after a step in which MedicalDoctors is retained in the multiple model (if any)?

273:

I'm still not clear which comparisons would involve non-nested models. If a single variable is added at each step, shouldn't the original model always be a subset of the new model (i.e. the new model only has one extra variable)? Consider adding an Appendix to more clearly illustrate the variable selection approach.

**Results**

-Does the analysis presented match the analysis plan?

-Are the results clearly and completely presented?

-Are the figures (Tables, Images) of sufficient quality for clarity?

Reviewer #3: In my opinion there remains significant problems with the presentation of results, and ambiguity as to why results for some variables/analyses are not presented.

Table 2 (Results):

I still don't understand why some variables presented in Tables 1 and 2 (Methods) are missing from Table 2 (Results) (likewise for Fig. 5, for the continuous variables), e.g.:

- interval from treatment to death

- race proportion

- illiteracy

- population growth rate

- average household income

- sewage

- PIB

- HDI

- health units (result described line 407)

- emergency beds (result described line 408)

Table 2 (Results):

- For categorical variables, in the "Univariate analysis" column there should only be odds ratios in cells *other* than the (base) level. This is currently the case for variables with 3 categories (e.g. Number of VL symptoms (%), Standardized VL incidence), where the odds ratio for the base level is replaced with "-". But for all the variables with 2 categories (Age, Gender, etc.) the odds ratio is placed on the line corresponding to the level indicated as "base", which makes interpretation ambiguous. E.g. For "Gender" the odds ratio is placed in the line corresponding to "Female (base)". Does this odds ratio represent the risk for females relative to males (as indicated by the placement of the OR), or males relative to females (as indicated by the designation "Female (base)")? Suggest consistent use of "-" for the base level cells for clarity.

351-360:

- Why present these results from the "initial global analysis" only for these 3 specific variables (age, onset to symptom delay, diagnostic criteria), and not all the other variables? It would be preferable to present all results in the same way. But at a minimum, if you choose to present only certain results for a given type of analysis, it should be clear to the reader how specifically you chose which results to present and which to omit.

- More broadly, I'd suggest omitting these 'global analysis' results altogether, as they're describing analyses that are basically just the inverse of the univariate models (i.e. they use patient outcome as an explanatory variable and the demographic/diagnostic variables as response).

358:

- These categories of diagnostic confirmation (parasitological test, IFAT serological testing, other serological testing) are missing from Table 1 (Methods)

365:

- Clarify the line "high correlation among several contextual variables (Figure 4), such as degree of urbanization and proportion of the population up to 5 years old, correlated to many other variables." Based on Fig 4, the variables Urbanization and Children<5 don't appear to be correlated above 85%.

367:

- Clarify whether this also accounts for negative correlations (e.g exclude |r| > 0.85) ?

385, 387: 

Should be referencing Figure 5 instead of 6 here I think

394:

The term "crude effect" is a bit confusing here, because the odds ratios in Table 2 are from univariate models that also include terms for FU and time, and adjust for age structure. They're not really "crude". Though based on the counts in Table 2, the crude effect for age (1911/12600)/(820/16506) = 3.05 is almost the same as the univariate odds ratio (3.01).

399:

change "discharge" to "discarded"

402-404:

Rephrase this result to clarify. If I understand correctly, this result is treating VL incidence as the response variable and patient outcome as the explanatory (like the "initial global analysis" on lines 351-360)? Suggest just presenting the univariate results here for consistency and clarity.

411:

I think this ref should be to Fig 5e (Emergency beds) instead. There are also references to Fig 6 that should be to Fig 5 (e.g. line 418).

Figs. 1b, 5:

I don't think "Effect" is a sufficient y-axis label here. Assuming these plots come from e.g. the plot() function in R used on a MGCV gam model object, the y-axis shows the predicted values of the y variable on the transformed scale (e.g. logit in this case), centered to a mean of zero based on the model intercept. A more accurate label would be something like "logit(Case fatality), centered".

**Conclusions**

-Are the conclusions supported by the data presented?

-Are the limitations of analysis clearly described?

-Do the authors discuss how these data can be helpful to advance our understanding of the topic under study?

-Is public health relevance addressed?

Reviewer #3: The latest revisions to add nuance to interpretations of results that differ between univariate and multiple analyses are a step in the right direction, but in my view there remain conclusions that are not justified by the methods/results, and a lot more nuance is needed for certain analyses.

The biggest problem relates to over-interpretation of analyses of variables that were truncated (arbitrarily in my opinion) prior to univariate analysis, particularly "# of Emergency beds" and "# of FHP units" (e.g. 580-584, 635-638).

E.g. The authors conclude...

"The link between access to diagnosis and health facilities and VL case-fatality is also suggested by the inverse association found in univariate analysis between availability of emergency beds, FHP teams and medical doctors and death from VL, which suggests that the robustness of the health system is directly related to the VL clinical outcome." (580-584)

and

"The variability in the installed structure of the public health system, expressed by the number of emergency beds and health multi-professional teams, in addition to the ability of the health system to recognize the disease, which is directly related to the regional incidence of VL, appear as additional markers of the risk of death." (635-638)

For "Emergency beds" and "FHP Units", the splines (Figs 5e, 5f) indicate an inverse relationship only in certain regions of the curve, and the authors have simply excluded later regions where fatality appears to increase with the variable of interest. This is not at all rigorous in my opinion.

The conclusion regarding the relationship between VL Incidence and fatality ("Lower VL incidence was also associated to VL case fatality, suggesting that unfamiliarity with the disease may delay appropriate medical management" (45-47)) should also be more nuanced given the relationship depicted in Fig 5d. E.g. the spline suggests that predicted case fatality is lower at an incidence of 60 (the first split point) compared to an incidence of 160 (the final split point).

With respect to the variable Medical doctors, the raw odds ratio for fatality above 1.5 doctors per capita compared to below is (1939 / 22156) / (792 / 6950) = 0.77 (with the CI entirely below 1), similar to the univariate analysis, which suggests that fatality generally declines with an increasing number of Medical doctors. But the spline relationship is very clearly showing the opposite effect. I understand that these are different modelling approaches, as the authors explained in their previous response, but these results are nonetheless *very* difficult to reconcile. E.g. the spline-predicted fatality above 1.5 doctors is higher at every point along its curve than the predicted fatality below 1.5 doctors.

Given that the authors make conclusions about this relationship (580-584), I would strongly suggest digging a bit further to explain this pattern (possibly in a Supplementary Appendix), and I reiterate that the most important fix would be to add the FU-specific data point to the figures (this applies to all spline figures). I show an example of how to do this with R code attached as a pdf to my review (I hope this is available to the authors).

Relatedly, the univariate analysis suggests higher case fatality for males compared to females (Table 2) whereas the multiple analysis suggests substantially *lower* case fatality for males (Table ). Again I understand that these are different analyses, but this difference warrants further investigation and explanation, rather than just affirming the finding that female gender is "a known risk factor".

**Editorial and Data Presentation Modifications?**

Reviewer #3: The reference numbering looks to be off again in places. E.g. at line 199, the reference [26,27] should instead of [27,28].

There is a Table 1 and 2 in the Methods, and another Table 1 and 2 in the Results. Normally tables are numbered sequentially through the entire manuscript.

Between submissions #2 and #3, the units in the axis labels for Figs 5a and 5c changed from "days" to "weeks", but the axis values didn't change. Just confirming that this was intentional? Also, Fig. 5h (Garbage collection) changed in a way that I don't understand. The range of the x-values (1.6-2.8) doesn't correspond with the units in Table 2 (Results)
---

## [Decision Letter · Decision Letter 3]

16 Jun 2021

Dear Dr Cota,

We are pleased to inform you that your manuscript 'Inequalities of visceral leishmaniasis case-fatality in Brazil: a multilevel modeling considering space, time, individual and contextual factors' has been provisionally accepted for publication in PLOS Neglected Tropical Diseases.

Best regards,

Alberto Novaes Ramos Jr

Associate Editor

Nadira Karunaweera

Deputy Editor

Reviewer's Responses to Questions

**Key Review Criteria Required for Acceptance?**

**Methods**

-Are the objectives of the study clearly articulated with a clear testable hypothesis stated?

-Is the study design appropriate to address the stated objectives?

-Is the population clearly described and appropriate for the hypothesis being tested?

-Is the sample size sufficient to ensure adequate power to address the hypothesis being tested?

-Were correct statistical analysis used to support conclusions?

-Are there concerns about ethical or regulatory requirements being met?

Reviewer #1: As stated previously, the methods are OK now.

Reviewer #2: (No Response)

Reviewer #4: Objectives are clearly described and statistical methods employed are adequate for achieving the proposed goals. There are no ethical concerns.

**Results**

-Does the analysis presented match the analysis plan?

-Are the results clearly and completely presented?

-Are the figures (Tables, Images) of sufficient quality for clarity?

Reviewer #1: As stated previously, the results are OK now.

Reviewer #2: (No Response)

Reviewer #4: Results are clearly presented and figures and tables adequate

**Conclusions**

-Are the conclusions supported by the data presented?

-Are the limitations of analysis clearly described?

-Do the authors discuss how these data can be helpful to advance our understanding of the topic under study?

-Is public health relevance addressed?

Reviewer #1: As stated previously, the conclusions are OK now.

Reviewer #2: (No Response)

Reviewer #4: Conclusions are well elaborated and the public health relevance adequately explored.

**Editorial and Data Presentation Modifications?**

Reviewer #1: Accept.

Reviewer #2: (No Response)

Reviewer #4: No modifictions are necessary.

**Summary and General Comments**

Reviewer #1: The article is ready to be published.

Reviewer #2: (No Response)

Reviewer #4: This manuscript has gone through three rounds of review, and I am delighted with this final version. Authors should be allowed to exert their latitude of decisions and choices regarding methods and manuscript format, as long as these choices are not inadequate, incorrect, or non-scientifically sound. Here, the authors employed an exciting and well-known approach for assessing the individuals and contextual effects of factors on the case-fatality rates of visceral leishmaniasis. The problem is of great relevance; the approach is somewhat innovative, the methods are adequate for the objectives, results are interesting, conclusions supported by the data and discussion advances on the relevance of the results for informing public health to deal with such a grave problem.

PLOS authors have the option to publish the peer review history of their article (what does this mean?). If published, this will include your full peer review and any attached files.

Reviewer #1: **Yes: **ccc

Reviewer #2: No

Reviewer #4: **Yes: **Guilherme Loureiro Werneck

---

## [Editor Report · Acceptance letter]

28 Jun 2021

Dear Dr Cota,

We are delighted to inform you that your manuscript, "Inequalities of visceral leishmaniasis case-fatality in Brazil: a multilevel modeling considering space, time, individual and contextual factors," has been formally accepted for publication in PLOS Neglected Tropical Diseases.

Best regards,

Shaden Kamhawi

co-Editor-in-Chief

Paul Brindley

co-Editor-in-Chief
